# Numerosity estimation of virtual humans as a digital-robotic marker for hallucinations in Parkinson's disease

Louis Albert [1], Jevita Potheegadoo[1], Bruno Herbelin[1], Fosco Bernasconi[1] & Olaf Blanke [1,2] ✉

Hallucinations are frequent non-motor symptoms in Parkinson's disease (PD) associated with dementia and higher mortality. Despite their high clinical relevance, current assessments of hallucinations are based on verbal self-reports and interviews that are limited by important biases. Here, we used virtual reality (VR), robotics, and digital online technology to quantify presence hallucination (vivid sensations that another person is nearby when no one is actually present and can neither be seen nor heard) in laboratory and home-based settings. We establish that elevated numerosity estimation of virtual human agents in VR is a digital marker for experimentally induced presence hallucinations in healthy participants, as confirmed across several control conditions and analyses. We translated the digital marker (numerosity estimation) to an online procedure that 170 PD patients carried out remotely at their homes, revealing that PD patients with disease-related presence hallucinations (but not control PD patients) showed higher numerosity estimation. Numerosity estimation enables quantitative monitoring of hallucinations, is an easy-to-use unobtrusive online method, reaching people far away from medical centers, translating neuroscientific findings using robotics and VR, to patients' homes without specific equipment or trained staff.

Parkinson's disease (PD) is the second most common neurodegenerative disease following Alzheimer's disease, affecting approximately 3% of the population over 65 years of age[1]. Although PD is defined primarily as a movement disorder (i.e. resting tremor, rigidity, bradykinesia), it is a heterogeneous disorder, also affecting several non-motor systems and manifesting in a wide variety of non-motor symptoms, including hallucinations[2]. Hallucinations in PD are highly prevalent (with approximately half of PD patients experiencing hallucinations) and can reach up to 70% during later stages of the disease[3]. Critically, hallucinations have been associated with a more severe form of PD with negative clinical outcomes including dementia, depression, early home placement, and a higher mortality[3–12].

Hallucinations in PD have been categorized into formed (well-structured or complex) visual hallucinations and so-called minor hallucinations, which include presence hallucinations and passage hallucinations, and visual illusions[5,13]. Visual hallucinations generally occur at the middle to late stage of the disease, and several studies have identified visual hallucinations as a risk factor for more rapid cognitive decline and dementia in PD[14–17]. However, because visual hallucinations occur at a more advanced stage of the disease, with cognitive decline already present, they are not suitable as an early marker of cognitive decline in PD. This differs for minor hallucinations, which are usually experienced at earlier stages of the disease[3,9], and can even precede parkinsonian motor symptoms, testifying to the importance to include them in detailed clinical evaluations[18]. Recent data show that minor hallucinations are not only the earliest hallucinations occurring in PD, but that they also share brain alterations with visual hallucinations[19,20] and are linked to more rapidly developing

[1]Laboratory of Cognitive Neuroscience, Neuro-X Institute, Faculty of Life Sciences, Swiss Federal Institute of Technology (EPFL), Geneva, Switzerland. [2]Department of Clinical Neurosciences, Faculty of Medicine, University of Geneva, Geneva, Switzerland. ✉e-mail: olaf.blanke@epfl.ch

cognitive deficits[8,21,22], underlining their potential role as an early marker for dementia[8,19,21,22].

Despite their clinical relevance, the diagnosis and investigation of hallucinations is hampered by difficulties to examine them in real-time and quantify them reliably. Hallucinations are subjective-private experiences[23] and self-reports are inadequate for precisely representing and quantifying hallucinations[23,24]. Yet, the current gold standard in medical practice for assessing type, frequency, and intensity of hallucinations is based on verbal self-reports and interviews of patients and interpretations by clinicians. However, verbal reports only capture some aspects of conscious experience and are prone to many biases (i.e., we dispose of a limited ability to accurately recall and report past conscious experiences, with an unknown reliability; many aspects of these conscious experiences cannot be accurately quantified in verbal reports; verbal reports are prone to the interviewer biases[23,25]). These limitations are further exacerbated, because patients reporting their hallucinations may be affected by lack of insight and/or fear of stigmatization, which often refrains patients from reporting them[26].

New procedures and methods have been developed to overcome some of these limitations, allowing a quantitative assessment of hallucinations and to investigate and induce hallucinations in controlled laboratory setting, in both healthy and clinical populations[24]. Of relevance for PD, a sensorimotor robotic procedure has been shown to induce repeatedly a specific and clinically relevant hallucination, presence hallucination (the vivid sensation that another person is nearby when no one is actually present and can neither be seen nor heard[27]), in healthy participants and in PD patients[24,28]. Such real-time induction of a clinically relevant hallucination in healthy participants has permitted to identify the brain mechanisms underlying these aberrant perceptions without the confounds present in clinical populations (i.e., co-morbidities not related to hallucinations)[22–24]. Moreover, the translation of the procedure and methods to patients with PD confirmed that robotically induced presence hallucinations are clinically relevant, because they shared key phenomenological aspects with PD patients' spontaneous presence hallucinations in daily life, and because PD patients with spontaneous presence hallucinations were more sensitive to the robotically induced presence hallucination procedure[24]. However, while the robotics-based approach enabled the investigation of presence hallucination in real-time within a controlled environment (i.e., overcoming several limitations of earlier hallucination research), the procedure still relied on explicit ratings (e.g., questionnaires), which can be sensitive to participant and experimenter biases[23,25,29], but may be overcome by implicit behavioral proxies[30], as applied to the sense of self-location[31,32], spatial thought[33,34], or agency judgements[35,36].

Here, we designed a novel and fully controlled behavioral numerosity task with visual virtual human agents (and with visual control objects: control task), using immersive Virtual Reality (VR) technology that we combined with our robotic procedure that allows to induce presence hallucinations in healthy participants[24] (study 1). We combined VR with the robotic system and determined whether the new task is an implicit, quantitative, and behavioral marker for robot-induced presence hallucinations, in a group of 28 healthy participants. Our findings reveal an overestimation bias for human stimuli that is robust and based on many repeated trials, which is observed in the condition that induces presence hallucinations and absent in the control task (object condition), establishing the overestimation of virtual human agents as an implicit behavioral marker for robot-induced presence hallucinations. Based on these results and our previous finding that PD patients with symptomatic presence hallucinations (compared to PD patients without those hallucinations) show heightened sensitivity to robotically induced presence hallucinations (independent of asynchronous stimulation)[22], we hypothesized that PD patients with symptomatic presence hallucinations (PD-PH) would show an overestimation for virtual human agents as compared to patients with PD but without hallucinations (PD-nH). To test this, we developed a home-based online numerosity task with virtual human agents and investigated a large group of 170 patients with PD at their home without robotic stimulation (study 2). Online data reveal a larger overestimation bias for virtual human agents in PD-PH patients with presence hallucinations as part of the disease as compared to control PD-nH patients, demonstrating an implicit digital online marker for early hallucinations in PD.

## Results

### Estimation of human stimuli using immersive Virtual Reality (study 1)

To test whether classical numerosity estimation effects observed for different visual stimuli such as dots[37–44], squares[45–47] or cartoon animals[48] can also be observed for more complex and ecologically valid stimuli such as virtual human agents in a room (as shown in a VR scenario), we developed a new immersive 3D VR paradigm. To maximize immersion and strengthen the ecological validity of our experiment, healthy participants were first immersed in a virtual reconstruction of the actual testing room, including the participants' actual location and orientation in the exact same room of our laboratory. Stimuli consisted in brief displays of the 3D VR environment, which contained a varying number of virtual human agents (Fig. 1; Supplementary Figs. 1, 2; Supplementary Movie 1) that were equally positioned in the virtual room, in front of the participant, at least at a distance of 1.75 virtual meters-, and within the near peripheral field of view. The 3D VR scene was displayed to participants on a head-mounted display (Oculus Rift CV1). Participants were asked to indicate the number of humans (human numerosity estimation task) they perceived in the room, as fast and as precise as possible. A control condition was also performed (number of objects (object numerosity estimation task); Supplementary Figs. 1, 2; Supplementary Movie 2).

To determine the lower bound of the estimation range of our human numerosity estimation task stimuli and select the range of presented numerosities in study 1, we conducted an online pilot study in 28 healthy participants (see Supplementary Note 1). In this online preliminary study, participants were asked to indicate the number of humans they perceived in flashed human stimuli over a broad range of numerosities (ranging from 1 to 24). This online preliminary study indicated the lower bound of the estimation range of our human numerosity estimation task stimuli to be 5. For all methodological aspects and detailed results of this online pilot study see Supplementary Note 1.

First, as predicted based on the previous literature[40,41,43], and in agreement with our preregistered hypothesis[49], our data show that numerosity estimation is modulated by the number of stimuli (humans or objects) presented in the virtual room (i.e., presented numerosity; $F(3, 2197) = 1946$; $p < 0.001$; main effect; Supplementary Table 3). In particular, we observed that numerosity estimation increases with the number of presented numerosities, independently of the type of stimulus (Supplementary Table 2; Supplementary Table 3). Second, and in agreement with our hypothesis[49], additional post-hoc analysis showed that participants mean numerosity estimation is significantly higher than the visually presented numerosity for the range of presented numerosity (5 to 8) (Fig. 2; Supplementary Table 2). This behavior is typical and has been observed with dots stimuli for numerosities just above the subitizing range[39,40,43]. The subitizing range corresponds to fast, accurate and confident number judgements, only observed for a low number of dots or items[41]. Above this range, the number of items can be either counted accurately but more slowly or estimated rapidly but with errors. These data confirm and extend two well-known effects from classical numerosity estimation (carried out with dots on 2D computer screens: numerosity main

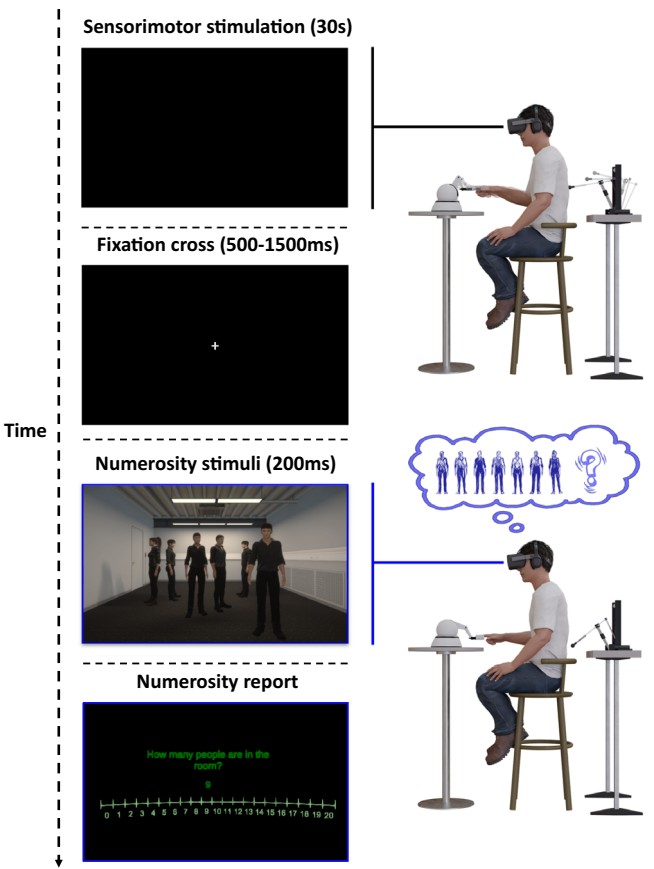

**Fig. 1 | Integrating sensorimotor robotic stimulation, virtual reality, and numerosity estimation task (study 1).** In each human numerosity estimation task trial, participants first manipulated the robotic system for 30 seconds (either in the asynchronous (500 ms delay; presence hallucination inducing condition) or the synchronous (0 ms delay)). This was followed by the appearance of a fixation cross (500–1500 ms), indicating to participants that they could stop moving the robotic system. Then, a scene containing a different number of people (range 5–8) was shown for 200 ms and participants had to estimate the number of people they saw. All visual stimuli were shown in immersive virtual reality on a head-mounted display (see methods for further detail). Please note that the numerosity stimuli were displayed in very dim lightning inside virtual reality (3D scenes), which is increased on the displayed material (2D picture) for presentation purpose.

effect, overestimation above the subitizing range)[39,40,43], that we also observed in our pilot online study (Supplementary Note 1). We here report them for the numerosity estimation of humans and control objects in immersive VR using a head-mounted display.

### Higher overestimation in human numerosity estimation task is associated with presence hallucinations (study 1)

To assess whether the magnitude of the human numerosity estimation task overestimation is a valid implicit measure for robot-induced presence hallucinations, our human numerosity estimation task stimuli and procedure were integrated with the robotic sensorimotor paradigm[24] that has been shown previously to induce presence hallucinations in healthy participants[22,28,50–52]. To induce presence hallucinations, participants were asked to perform repetitive movements to operate a robot placed in front of them, which was combined with a back robot providing tactile feedback to the participants' backs with a delay of 500 ms (asynchronous sensorimotor stimulation) (Fig. 1). A second sensorimotor condition (synchronous sensorimotor stimulation) served as a control condition (participants performed the same repetitive movements to operate the front robot and received the same tactile feedback on their backs, and with the same spatial

conflict, but without the additional 500 ms temporal delay of the asynchronous condition). On a trial-by-trial basis, participants performed the human numerosity estimation task (using the immersive VR procedure) immediately after each sensorimotor stimulation phase of 30 seconds (i.e., either asynchronous or synchronous sensorimotor stimulation), for a total of 40 human numerosity estimation task trials (for protocol see Fig. 1 and Supplementary Fig. 1). To reinforce the context of our human numerosity estimation task, as a habituation phase prior each task, participants were immersed in the virtual room for one minute. During this habituation phase several virtual human agents moved and were seen discussing among themselves in the virtual environment. We tested whether: i) sensorimotor stimulation is associated with changes in human numerosity estimation and, ii) in particular, if the magnitude of human numerosity estimation task overestimation can be used as an implicit marker of presence hallucinations in the asynchronous versus synchronous sensorimotor condition. That is, we hypothesize that the robot-induced invisible presence would increase the numerosity estimation bias of participants when exposed to the virtual human agents displayed on the head-mounted display. Critically, we predicted that this effect should be larger in the asynchronous versus synchronous sensorimotor condition and it would be absent in our control object numerosity estimation task (where instead of virtual human agents, control objects are shown in the same virtual room, at the same positions and orientations, and for the same numerosities) (Supplementary Fig. 1; Supplementary Fig. 2). In agreement with our preregistered hypothesis[49], we observed an overestimation in the presence hallucination inducing asynchronous sensorimotor condition that was specific to virtual human agents (i.e., absent for objects). Indeed, our results show that numerosity estimation is significantly ($F(1, 2197) = 11.5$; $p < 0.001$; Interaction; Supplementary Table 3) modulated by the synchrony of sensorimotor stimulation and the type of stimuli (virtual human agents vs. objects). Critically, and in agreement with our preregistered hypothesis[49], post-hoc comparisons showed that sensorimotor robotic stimulation significantly modulates human numerosity estimation ($t(2197) = -2.9$; $p = 0.003$; Supplementary Table 3; effect size $= -0.18$ (95% confidence interval $= [-0.29; -0.06]$)) (Fig. 3a; Fig. 3b) and that the presence hallucination inducing asynchronous condition induces a stronger human numerosity overestimation bias than the synchronous control condition. Moreover, this effect was only present when estimating the number of virtual human agents in the virtual room, as it was absent for objects ($t(2197) = 1.87$; $p = 0.06$; Supplementary Table 3; effect size $= 0.11$ (95% confidence interval $= [-0.01; 0.23]$)) (Fig. 3c; Supplementary Fig. 6). Collectively, these data show that the human numerosity overestimation depends on sensorimotor stimulation, that it is larger in the asynchronous sensorimotor condition versus synchronous sensorimotor control condition and that this modulation is not observed for the object numerosity estimation task. These findings reveal an overestimation bias for human stimuli in the presence hallucination inducing asynchronous sensorimotor condition, suggesting that human numerosity estimation task is an implicit behavioral marker or proxy for robot-induced presence hallucinations.

Additional analysis revealed no significant differences in response times between human and object numerosity estimation tasks (i.e., type of stimuli; $F(1,2197) = 0.73$; $p = 0.39$; no main effect plus no interaction; Supplementary Table 4) (Supplementary Fig. 4; Supplementary Fig. 5; Supplementary Fig. 7; Supplementary Fig. 8). This analysis also indicated no effect of robotic sensorimotor stimulation on response time (i.e., type of robotic sensorimotor stimulation; $F(1,2197) = 3.01$; $p = 0.08$; no main effect, no interaction; Supplementary Table 4), suggesting that the different robotic sensorimotor stimulation conditions did not affect numerosity estimation task difficulty nor alertness. This is supported by the fact that the numerosity estimation task is not performed during, but just after the robotic stimulation.

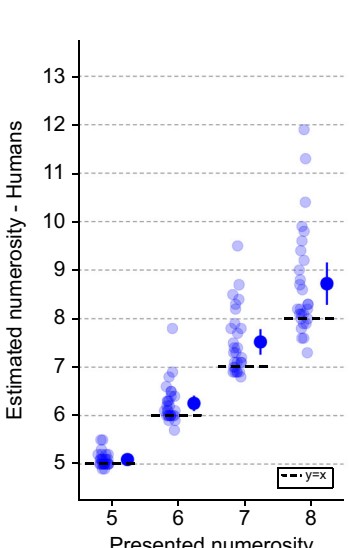

**Fig. 2 | Numerosity estimation (study 1).** General numerosity estimation performance for each tested numerosity in (**a**) the human numerosity estimation task and (**b**) the object numerosity estimation task (study 1). Each dot indicates the individual numerosity estimation task mean estimate for the corresponding presented numerosity. The dots with the bar on the right sides indicate the in-between subject mean for each presented numerosity. Note the general overestimation bias in the human numerosity estimation task and the object numerosity estimation task. Error bar represents 95% confidence interval. $n = 28$ healthy participants. Source data are provided as a Source Data file.

## Presence hallucinations are stronger in asynchronous versus synchronous sensorimotor condition (study 1)

To ensure the successful induction of presence hallucinations during the robotic sensorimotor stimulation, we additionally administered, at the beginning of the experiment and prior to the numerosity estimation task, a previously used comprehensive questionnaire about presence hallucinations (Supplementary Table 1, Supplementary Fig. 2). In line with previous results observed with the robotic sensorimotor paradigm[22,28,50–52], the present participants reported higher presence hallucinations ratings in the asynchronous sensorimotor condition (mean = 2.29, SD = 1.96) compared to the synchronous sensorimotor condition (mean = 1.50, SD = 1.99) ($\chi^2$ (1, $N = 28$) = 12.00, $p < 0.001$; effect size = -0.61 (95% confidence interval = [−1.01; −0.20])) (Fig. 4a; Supplementary Fig. 3; Supplementary Table 1). Other robot-induced bodily experiences (illusory self-touch, passivity experience and loss of agency) were also compatible with previous findings[22,28,51,52] (Supplementary Table 1, Supplementary Fig. 3).

## Overestimation of visual virtual human agents correlates with presence hallucination magnitude (study 1)

To corroborate that the human numerosity estimation task is an implicit marker for presence hallucinations, we conducted additional correlation analysis between presence hallucination ratings and the magnitude of overestimation in the human numerosity estimation task. This analysis revealed that the effect of sensorimotor stimulation on the human numerosity estimation task was partially mediated via presence hallucination ratings (Fig. 4b, Supplementary Note 2). This result indicates that the stronger our participants experienced presence hallucinations during the asynchronous (versus synchronous) sensorimotor condition, the higher was their overestimation of virtual human agents shown in the virtual room (human numerosity estimation task). This finding was absent between presence hallucination ratings and the object numerosity estimation task: the same analysis for the object numerosity estimation task during sensorimotor stimulation did not reveal any association between presence hallucinations and the object numerosity overestimation (Supplementary Note 3), further confirming the selectivity of the human numerosity

overestimation as a marker for the proneness to experience a presence hallucination.

## Experiencing an invisible presence increases visual overestimation bias for numerosity estimations of visual virtual human agents (summary of study 1)

The main finding from Study 1 is that we characterize human numerosity overestimation as a hallucination marker, showing that asynchronous sensorimotor stimulation is associated with an increase in human numerosity estimation. Critically, this effect was observed (1) when comparing the presence hallucination-inducing asynchronous sensorimotor condition with the synchronous sensorimotor control condition, and (2) was absent in the object numerosity estimation task. Observing that (3) the magnitude of the human numerosity estimation task bias, but not the object numerosity estimation task bias, correlates with presence hallucination ratings, further links the human numerosity overestimation with presence hallucinations. Accordingly, we argue that a robotically induced mental state (i.e., the induction of a hallucinatory invisible percept) systematically modulates performance in a visual task (human numerosity estimation task) by increasing the magnitude of the number of seen humans. The human numerosity estimation task is a robust marker for a specific and clinically relevant hallucination, presence hallucination, and is elicited in a controlled laboratory setting without relying on verbal ratings. The human numerosity estimation task can be repeated as many times as needed, across control conditions, with full control over the presented stimuli. Integrated into fully automatized and a virtual training and test scenario, the human numerosity estimation task thereby overcomes many limitations of previous hallucination research[23,24].

## Human numerosity estimation task in patients with Parkinson's disease (study 2)

These results show that human numerosity overestimation indexes experimentally induced presence hallucinations in healthy participants. Would this quantitative, digital, and implicit maker for presence hallucinations also extend to neurological patients, who experience presence hallucinations as part of their disease? Would the

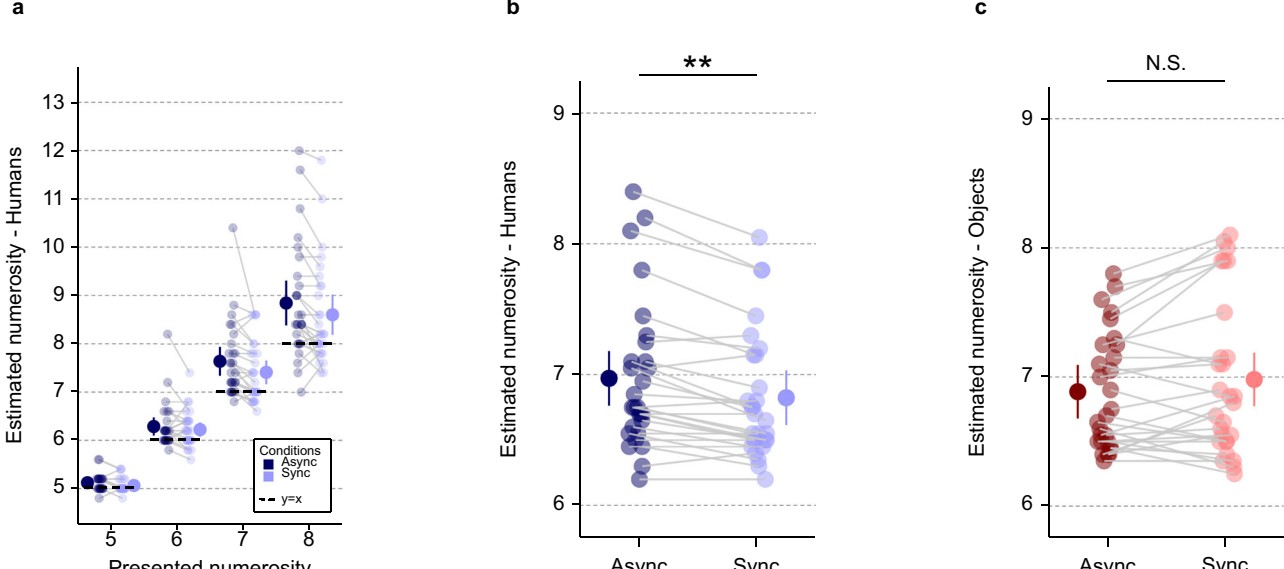

**Fig. 3 | Human and object numerosity estimation tasks as a function of sensorimotor stimulation (synchronous, asynchronous) (study 1). a** Task performance is shown for each presented numerosity in the human numerosity estimation task, for the asynchronous (dark blue) and synchronous (light blue) sensorimotor stimulation. Each linked pair of dots indicates the individual human numerosity estimation task mean estimate for the corresponding numerosity in asynchronous (dark blue) and synchronous (light blue) sensorimotor stimulation. The dots with the bar on the left and right sides indicate the mixed-effects linear regression between asynchronous (dark blue) and synchronous (light blue) sensorimotor stimulation for each tested numerosity. **b** Human numerosity estimation task (asynchronous (dark blue) versus synchronous (light blue) sensorimotor stimulation). Sensorimotor stimulation significantly modulates human numerosity estimation (t(2197) = −2.9; p = 0.003; effect size = −0.18 (95% confidence interval = [−0.29; −0.06])). Each linked pair of dots indicates the individual human

numerosity estimation task mean estimate in asynchronous (dark blue) and synchronous (light blue) sensorimotor stimulation. The dots with the bar on the left and right sides indicate the mixed-effects linear regression between asynchronous (dark blue) and synchronous (light blue) sensorimotor stimulation. **c** Object numerosity estimation (asynchronous (dark red) versus synchronous (light red) sensorimotor stimulation). Sensorimotor stimulation does not significantly modulates object numerosity estimation (t(2197) = 1.87; p = 0.06; effect size = 0.11 (95% confidence interval = [−0.01; 0.23])). Each linked pair of dots indicates the individual object numerosity estimation task mean estimate in asynchronous (dark red) and synchronous (light red) sensorimotor stimulation. The dots with the bar on the left and right sides indicate the mixed-effects linear regression between asynchronous (dark red) and synchronous (light red) sensorimotor stimulation. Error bar represents 95% confidence interval. **P ≤ 0.01. N.S., not significant. n = 28 healthy participants. Source data are provided as a Source Data file.

performance of PD patients with presence hallucinations as part of the disease be characterized by an increase in human numerosity estimation, compared to PD patients without presence hallucinations, even without any robotic stimulation? Recent work adapting the robotic presence hallucination induction paradigm to PD indicated that PD patients experiencing symptomatic presence hallucinations (PD-PH) had a six-fold higher sensitivity to the robotic sensorimotor procedure as compared to PD patients who never had presence hallucination (PD-nPH), suggesting that experiencing symptomatic presence hallucinations results in a bias in experiencing robot-induced presence hallucinations[22]. In combination with the results from study 1, these clinical data suggest that (1) PD-PH patients may have a bias in the human numerosity estimation task, that (2) this bias should exist without being exposed to sensorimotor robotic stimulation, and that (3) such a human numerosity estimation bias should be larger than the one in PD patients without such hallucinations (PD-nH).

In study 2, we aimed to test human numerosity estimation online and at home in a large cohort of PD patients in order to investigate whether the human numerosity estimation task reveals the occurrence of presence hallucinations in patients with disease-related spontaneous hallucinations in PD. This would be an important achievement, as it would allow to test patients directly at home, again without the biases associated with explicit verbal questionnaire evaluations or interviews[23,25]. Easy-to-use and unobtrusive online methods have gained momentum, by showing that large patient groups can be sampled, demonstrating the feasibility and validity of online methods and opening new perspectives towards better diagnostics and monitoring of diseases such as PD[53–57]. Digital online human numerosity estimation task testing also facilitates reaching people living far away

from medical centers[58], in low-income countries, without specific equipment (i.e., robotics, VR) and trained staff to perform hallucination testing.

We thus designed an online human numerosity estimation task by adapting the method used in study 1 (immersive 3D VR paradigm integrated with robotics) to a web-based 2D task for human and object (control) numerosity estimation tasks without either VR or robotic stimulation. PD patients performed this web-based digital task at home, by themselves on their own personal computer or tablet. In addition to the human and object numerosity estimation tasks data, we also acquired online questionnaire data about a range of demographical characteristics about the participants and about the occurrence of their hallucinations in daily life.

### Demographical and clinical data (study 2)
A total of 170 PD patients participated in our online experiment (https://go.epfl.ch/alpsn5, testing was carried out from August 2021 to June 2022) (Supplementary Fig. 14; Supplementary Note 4). From these, we included a total of 118 PD patients in the current analysis: 63 PD patients with presence hallucinations (PD-PH) and 55 PD patients without any hallucinations (PD-nH) (see Methods). Analysis of demographic data did not show any significant differences in gender, age, disease duration, nor medication (Levodopa equivalent daily dose) between the two patient groups (i.e., PD-PH vs. PD-nH; Table 1).

### Numerosity estimation for 2D human stimuli in a home-based online setting using participants' personal computer (study 2)
To test numerosity estimation for different numerosities of humans and to evaluate whether classical numerosity estimation effects can

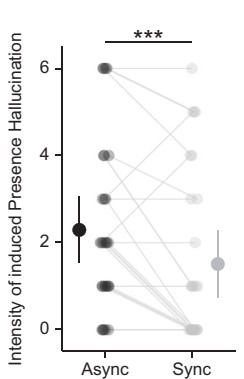

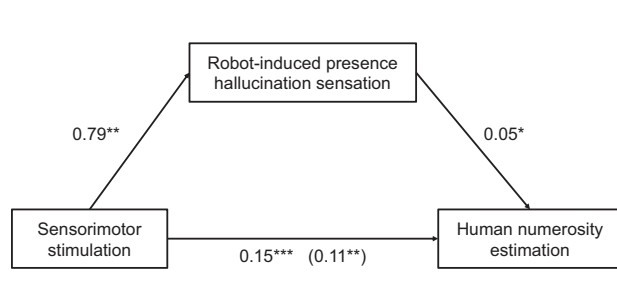

**Fig. 4 | Robot-induced presence hallucination and its link to human numerosity estimation (study 1). a** Robot-induced presence hallucinations assessment ratings (asynchronous versus synchronous sensorimotor stimulation). Participants reported higher presence hallucinations ratings in the asynchronous sensorimotor condition (mean = 2.29, SD = 1.96) compared to the synchronous sensorimotor condition (mean = 1.50, SD = 1.99) ($\chi^2$ (1, N = 28) = 12.00, p = 0.0005; effect size = −0.61 (95% confidence interval = [−1.01; −0.20])). Each linked pair of dots indicates the individual mean rating of robot-induced presence hallucination (asynchronous (dark grey) and synchronous (light grey) sensorimotor stimulation. The dots with the bar on the left and right sides indicate the mixed-effects linear regression between asynchronous (dark grey) and synchronous (light gray) sensorimotor stimulation. Error bar represents 95% confidence interval. **b** Results of causal mediation analysis. The effect of robotic sensorimotor stimulation (synchronous or asynchronous) on human numerosity estimation was partially mediated via robot-induced presence hallucination question rating. The regression coefficient between robotic sensorimotor stimulation (synchronous or asynchronous) and human numerosity estimation was significant (($F_{(1,27)}$ = 26.05; p = 2.3e−5)). The regression coefficient between robot-induced presence hallucination question rating and human numerosity estimation was significant ($F_{(1,27)}$ = 10.44; p = 0.003). The indirect effect of robotic sensorimotor stimulation (synchronous or asynchronous) on human numerosity estimation via robot-induced presence hallucination question rating was 0.04. The significance of the indirect effect was tested using bootstrapping procedures (1000 samples), and the 95% confidence interval was computed by determining the indirect effects at the 2.5th and 97.5th percentiles. The indirect effect was significant (p = 0.01; 95% confidence interval = [0.006; 0.08]). *$P \leq 0.05$; **$P \leq 0.01$; ***$P \leq 0.001$. n = 28 healthy participants. Source data are provided as a Source Data file.

also be observed for human stimuli on a 2D screen (as observed previously for visual dots)[37–44] and for 3D virtual human agents (i.e., study 1), we converted our 3D VR stimuli, used in the human and object numerosity estimation tasks of study 1, into 2D stimuli that we displayed on participants' computer screen or tablet at home. We presented a varying number of virtual human agents (Supplementary Fig. 2) that were equally positioned across the depicted room, characterized by the same configurations as used for the VR stimuli of study 1. Following a screen calibration procedure to control the size of the displayed stimuli, each participant performed the online numerosity estimation task. Based on previous numerosity estimation work, participants were asked to indicate the number of humans (online human numerosity estimation task) they perceived in the virtual room, as fast and as precise as possible (Fig. 5). For the control condition (online object numerosity estimation task) the virtual human agents were replaced with objects (boxes; online object numerosity estimation task, Supplementary Fig. 2), positioned in the same way as the virtual human agents and in the same virtual room. In the control

object numerosity estimation task, participants were asked to indicate the number of objects they perceived in the virtual room.

First, as predicted based on previous literature[40,41,43] and on the data of study 1, online data show that numerosity estimation is modulated by the number of stimuli (humans or objects) presented in the virtual room (i.e., presented numerosity; $F_{(3, 9084)}$ = 2055; p < 0.001; main effect; Supplementary Table 5). In particular, we observed that numerosity estimation increases with the number of presented numerosities, independently of the type of stimulus (Supplementary Table 5; Supplementary Table 9; Supplementary Note 5; Supplementary Table 10). Second, additional post-hoc analysis showed that participants mean numerosity estimation is significantly higher than the presented numerosity in the range of presented numerosities (5 to 8) (Fig. 6; Supplementary Table 5). This is a typical behavior observed with dots stimuli for numerosity just above the subitizing range[39,40,43]. These data extend two well-known known effects from numerosity estimation studies that have been carried out in research laboratories in young healthy participants (numerosity main effect, overestimation just above the subitizing range) to PD patients, who carried out the task on their personal computer or tablet at home.

### Presence hallucinations in patients with Parkinson's disease are associated with higher overestimation in human numerosity estimation task (study 2)

Our main research questions were whether the occurrence of presence hallucinations in PD (PD-PH group) is associated with changes in online human numerosity estimation task and, in particular, if spontaneous presence hallucinations are associated with overestimation in the online human numerosity estimation task, as observed in healthy participants after robot-induced presence hallucination (study 1). Additionally, based on the findings of study 1, we predicted that such an effect would be absent in the online control object numerosity estimation task (Supplementary Fig. 2). In agreement with our hypothesis, we observed an overestimation in the PD-PH group as

### Table 1 | Clinical variables (study 2)

|  | PD-nH (N = 55) | PD-PH (N = 63) | p value |
|---|---|---|---|
| **Gender** | 27 (M) | 29 (M) | 0.88 ($\chi^2$) |
| **Age (years)** | 66.11 ± 8.14 | 64.59 ± 7.57 | 0.30 |
| **PD duration (years)** | 6.18 ± 5.26 | 6.42 ± 4.95 | 0.80 |
| **LEDD (mg/day)** | 276.56 ± 317.20 (N = 53) | 318.49 ± 315.11 (N = 61) | 0.48 |

Clinical variables of PD-PH and PD-nH included in the numerosity estimation task analysis. Of the 170 patients with PD, 118 patients of interest to answer our research question (PD-PH (n = 63) and PD-nH (n = 55)) were kept for the analysis of the human and object numerosity estimation tasks (see methods for further detail). The Supplementary Table shows the mean and standard deviation for several clinical and demographic variables. There was no significant difference between groups in terms of gender ($\chi^2$ (1, N = 118) = 0.022, p = 0.88), age (t(111.1) = 1.05; p = 0.30), PD duration (t(111.7) = −0.26; p = 0.80) and LEDD (t(109.6) = −0.71; p = 0.48). *PD* Parkinson's Disease, *PD-PH* Parkinson's Disease patients with Presence Hallucination, *PD-nH* Parkinson's Disease patients with no Hallucination, *LEDD* Levodopa Equivalent Daily Dose.

compared to the PD-nH group. Moreover, overestimation was only present for the online human numerosity estimation task, but absent for the online object numerosity estimation task. Indeed, our results show that online numerosity estimation is significantly (F(1, 9085) = 53.81; $p$ < 0.001; interaction; Supplementary Table 9) modulated by the occurrence of presence hallucinations (PD-PH vs PD-nH) and by the type of stimulus (virtual human agents vs objects). Critically, and in agreement with our hypothesis, post-hoc comparisons

showed that the occurrence of presence hallucinations significantly modulates online human numerosity estimation task (t(122) = −3.16; $p$ = 0.002; Supplementary Table 9; effect size = −0.41 (95% confidence interval = [−0.67; −0.15])) (Fig. 7a, b), with PD patients who reported presence hallucinations showing stronger online human numerosity overestimation. Moreover, this effect was only present when estimating the number of humans in the virtual room and was absent for control online object numerosity estimation task: further post-hoc analyses did not reveal any statistical difference in online object numerosity estimation task between PD groups (PD-PH vs PD-nH) (t(122) = -0.59; $p$ = 0.42; Supplementary Table 9; effect size = −0.10 (95% confidence interval = [-0.36; 0.15]))) (Fig. 7c and Supplementary Fig. 11). These data show that PD patients with presence hallucinations show a human numerosity overestimation bias that can be measured online at the patient's home. We also corroborate the specificity of the overestimation bias for human stimuli because the effect was absent in the online object numerosity estimation task, showing that the online human numerosity estimation task is an implicit digital online marker for presence hallucinations in PD patients.

Additional analysis revealed no statistical differences in response times between the two PD groups (i.e., PD group; F(1,116) = 0.58; $p$ = 0.45; no main effect nor any interaction; Supplementary Table 11) (Supplementary Fig. 9; Supplementary Fig. 10; Supplementary Fig. 12; Supplementary Fig. 13), suggesting that task difficulty (online human and object numerosity estimation tasks) did not differ between both PD groups (PD-nH and PD-PH) (Supplementary Note 6).

### Online numerosity estimations of visual virtual human agents reveals presence hallucinations in Parkinson's disease (summary of study 2)

We successfully adapted our previously developed method of study 1 (immersive VR human and object numerosity estimation tasks; robotic sensorimotor stimulation) into a digital procedure that is fully online, does not require any robotic stimulation, and engaged PD patients at their homes performing the task on their personal computer or tablet (the dropout rate during the numerosity estimation task was only 7% (Supplementary Note 7); reasons for participant drop-out are

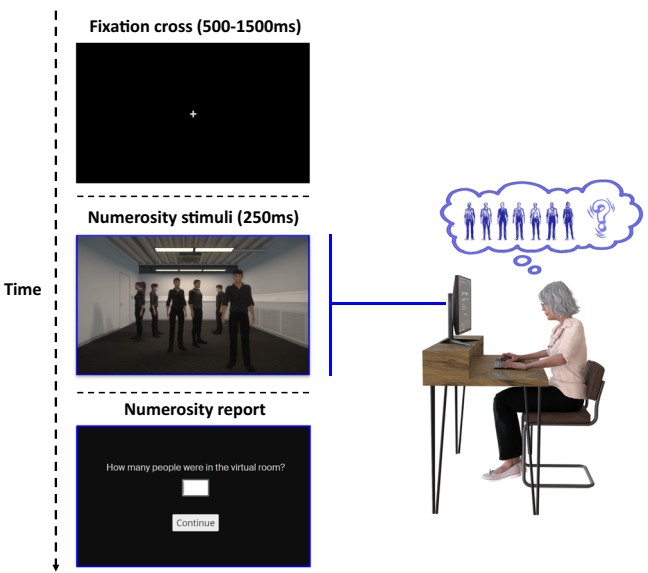

**Fig. 5 | Online human numerosity estimation task (study 2).** A single online human numerosity estimation task trial is shown. That started with the appearance of a fixation cross (500–1500 ms). After that, a scene containing different number of people (range 5–8) was shown for 250 ms and PD patients were asked to estimate the number of people that they saw. PD patients performed this web-based digital task at home, on their personal computer or tablet. PD Parkinson's Disease.

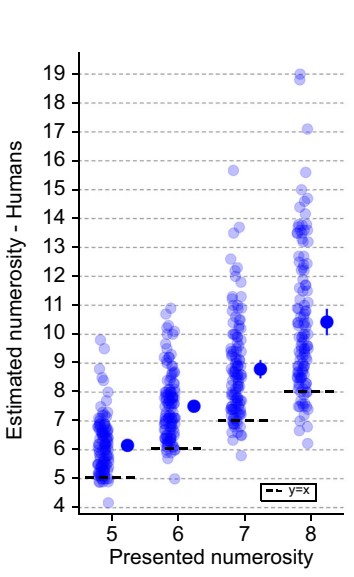

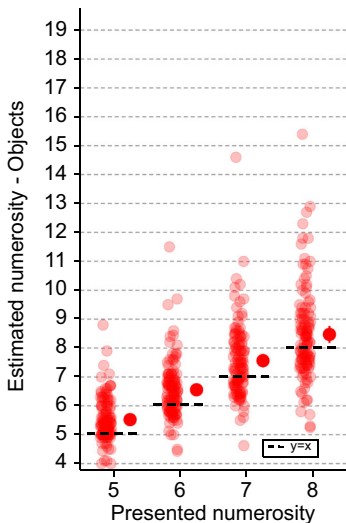

**Fig. 6 | Numerosity estimation (study 2).** General numerosity estimation performance for each tested numerosity in the **a** human numerosity estimation task and **b** object numerosity estimation task (study 2). Each dot indicates the individual human numerosity estimation task mean estimate at the corresponding tested numerosity. The dots with the bar on the right sides indicate the in-between subject mean at each presented numerosity. Note the general overestimation bias in human numerosity estimation task and object numerosity estimation task. The error bar represents 95% confidence interval. $n$ = 118 patients with PD. Source data are provided as a Source Data file. PD Parkinson's Disease.

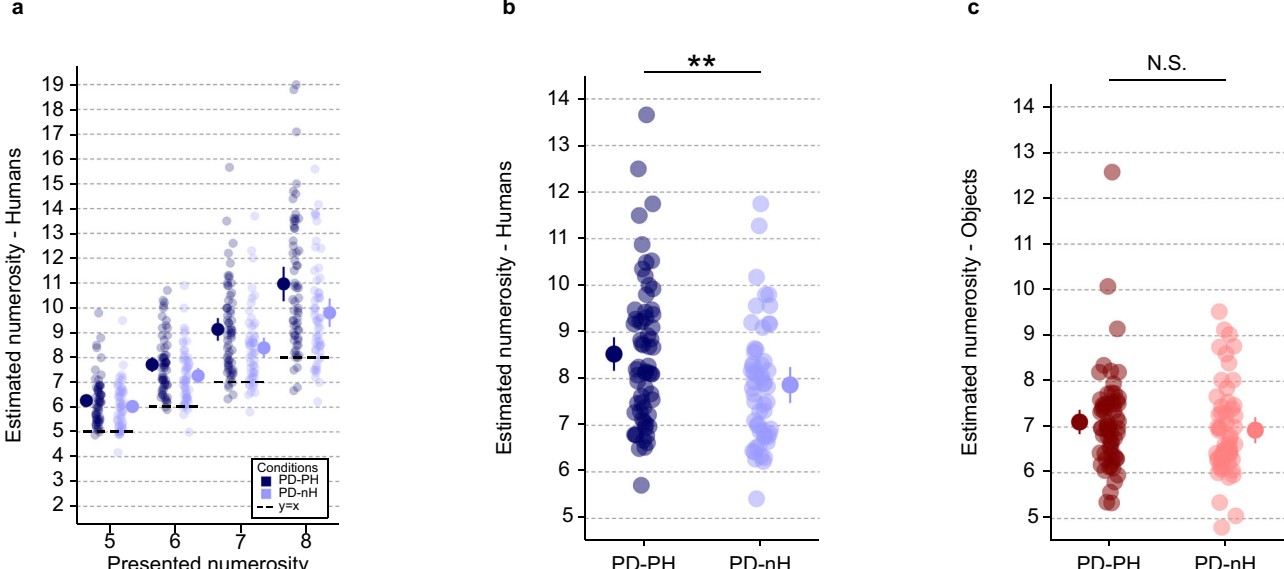

**Fig. 7 | Human and object numerosity estimation tasks for both PD patient groups (PD-PH and PD-nH) (study 2). a** Performance is shown in PD patients for each tested numerosity in the human numerosity estimation task for PD-PH (dark blue) and PD-nH (light blue) separately. Each dot indicates the individual human numerosity estimation task mean estimate for the tested numerosity (PD-PH (dark blue) and PD-nH (light blue)). The dots with the bar on the left and right sides indicate the mixed-effects linear regression between PD-PH (dark blue) and PD-nH (light blue) at each presented numerosity. **b** Human numerosity estimation task in PD patients (PD-PH vs PD-nH). The occurrence of presence hallucinations significantly modulates online human numerosity estimation task (t(122) = −3.16; p = 0.002; effect size = −0.41 (95% confidence interval = [−0.67; −0.15])). Each dot indicates the individual human numerosity estimation task mean estimate (PD-PH (dark blue) and PD-nH (light blue)). The dots with the bar on the left and right sides indicate the mixed-effects linear regression between PD-PH (dark blue) and PD-nH (light blue). **c** Object numerosity estimation task in PD patients (PD-PH vs PD-nH). No statistical difference in online object numerosity estimation task was observed between PD groups (PD-PH vs PD-nH) (t(122) = −0.59; p = 0.42; effect size = −0.10 (95% confidence interval = [−0.36; 0.15])). Each dot indicates the individual object numerosity estimation task mean estimate (PD-PH (dark red) and PD-nH (light red)). The dots with the bar on the left and right sides indicate the mixed-effects linear regression between PD-PH (dark red) and PD-nH (light red). Error bar represents 95% confidence interval. **P ≤ 0.01. n = 118 patients with PD (63 PD-PH & 55 PD-nH). Source data are provided as a Source Data file. PD Parkinson's Disease; PD-PH Parkinson's Disease patients with Presence Hallucination, PD-nH Parkinson's Disease patients with no Hallucination.

indicated in Supplementary Note 8). Applying this new online procedure, we, critically, show that the online performance of PD-PH patients is characterized by a stronger online human numerosity overestimation bias (as compared to PD patients without hallucinations; PD-nH), thereby linking clinical presence hallucinations to online human numerosity overestimation. This effect was absent in the online object numerosity estimation task and does not depend on clinical covariates such as age, gender, disease duration, or first affected side, corroborating data from study 1, extending them to a large clinical cohort of PD patients, and underlining human numerosity overestimation as a digital marker of presence hallucinations in PD.

## Discussion

We developed a new paradigm (human numerosity estimation task) to implicitly assess a clinically relevant hallucination: presence hallucination. In a first study in healthy participants, we tested our paradigm by merging immersive VR with a robotic platform able to experimentally induce presence hallucinations in a controlled manner and allowing for the real-time investigation of presence hallucinations[22,24,28]. Our results show that robot-induced presence hallucinations are associated with a selective overestimation for virtual human agents (human numerosity estimation task), but not for control objects (object numerosity estimation task). These results confirm our hypothesis that the human numerosity estimation task is a robust digital marker for the specific hallucinatory mental state of a presence hallucination. Because hallucinations are frequent and clinically relevant phenomena in PD (introduction and see below), in a second study, we adapted our numerosity estimation task to a web-based digital test and investigated 170 PD patients remotely at their homes. Translating our procedure to an online assessment, we validate online human

numerosity overestimation as a digital marker for presence hallucinations occurring as a symptom in PD. Using several controls, we rule out the possibility that these online human numerosity estimation task effects are confounded by clinical variables or task demands.

In study 1, we characterize human numerosity overestimation as a hallucination marker by showing that the presence hallucination inducing robotic sensorimotor condition is associated with a higher human numerosity estimation task bias (Fig. 3a, b), linking a robot-induced mental state with human numerosity overestimation. Importantly, this effect is absent in the object numerosity estimation task (Fig. 3c). We further report a correlation between the magnitude of the human numerosity overestimation and presence hallucination ratings (Fig. 4b), corroborating the link between human numerosity overestimation with presence hallucinations. These results suggest that the robot-induced presence hallucination state systematically modulates the performance in the human numerosity estimation task by increasing the magnitude of the number of people perceived. Compared to previous implicit measures used to assess presence hallucinations (e.g. drift in self-location[28], numerosity of actual people close by[28]), the present human numerosity estimation task has several advantages. The presented human stimuli are fully controlled, are tested for different and larger numerosities, and based on many repeated trials. Critically, the present procedure encompasses a carefully matched control condition with non-human objects (object numerosity estimation task), and we found no asynchrony-dependent overestimation when our healthy participants judged the number of non-human stimuli (object numerosity estimation task), controlling for task demand. Finally, the human numerosity estimation task can be repeated numerous time for all experimental stimuli and conditions and is fully orthogonal to the robotic sensorimotor manipulation,

overcoming limitations of previous hallucination research[23,24] and indexing a specific and clinically relevant hallucination in PD: presence hallucination.

In study 2, we successfully translated the human numerosity estimation task measurements to an online assessment of hallucinations performed by PD patients, who carried out the task on their personal computer or tablet at home. Presence hallucinations are an important symptom in Parkinson's disease, as they are usually experienced at early stages of the disease[3,9], may even precede Parkinsonian motor symptoms[18], and are linked to more rapidly advancing cognitive decline[8]. The present data show that PD patients with disease-related presence hallucinations in their daily life have a stronger online human numerosity estimation task bias and overestimation, as compared to PD patients without any hallucinations (Fig. 7a, b). This effect is absent for the online object numerosity estimation task assessment (Fig. 7c), and does not depend on any of the acquired clinical covariates (i.e., age, gender, disease duration, first affected side). The online human numerosity estimation task bias in PD-PH (study 2) thus extends the human numerosity estimation task bias, as induced by asynchronous robotic stimulation in healthy individuals (study 1), to patients with PD. We argue that the online human numerosity estimation task is a robust and quantitative digital marker for presence hallucinations in PD and, potentially, for other hallucinations and for cognitive decline, because presence hallucinations share brain alterations with visual hallucinations[19,20] and have been linked to cognitive impairment[22] and more rapid cognitive decline[8,21]. Compared to current standard methods used to assess hallucinations in the clinic (based on verbal self-reports and interviews of patients and interpretations by clinicians), which are associated with well-known biases, our implicit online human numerosity estimation task assessment overcomes several methodological limitations such as participant and experiment biases and underreporting due to fear of stigmatization[23–25]. Thus, the present online human numerosity estimation task measure constitutes a new promising digital marker for the quantitative assessment and monitoring of a more severe form of PD, associated with hallucinations as well as cognitive decline[59,60]. The present findings add to the recent upsurge of the impact of digital health technologies in PD[55,61]. These studies collected digital measures focusing on motor symptoms and their fluctuations (i.e., ~70% of studies on motor function vs. ~10% on cognitive function)[59], whereas the present human numerosity estimation task findings constitutes an online digital health marker for hallucinations in PD. Easy to use and unobtrusive online methods allow the investigation of much larger patient groups and facilitate longitudinal sampling, enabling better diagnostics and monitoring[53–56]. Clinic-based versus home-based evaluations often provide only a single snapshot of patient performance and this may not properly reflect a patient's performance in daily life at home[53,54,62]. The present digital online human numerosity estimation task testing has the additional advantage of reaching people living far away from medical centers, in low income countries, without requiring any specific equipment or trained staff[55,58].

Robot-induced presence hallucinations have been associated with a perturbation of sensorimotor self-related prediction signals[24,28,63]. That is, our robotic system creates a spatial mismatch between participants' right arm movements and their sensory consequences (tactile feedback) on the back. Combined with a temporal mismatch (asynchronous sensorimotor condition), this mismatch is resolved by participants perceiving the touch as originating from an external source (another agent, the hallucinated presence). When testing human numerosity estimation, we argue that the experimentally induced 'presence' modulates the number of humans estimated to be in the virtual room, as if the 'presence' is added to the seen humans. Robot-induced presence hallucinations in healthy participants have been associated with activation of the primary motor cortex, somatosensory cortex, premotor cortex and superior parietal lobule, bilateral supplementary motor area, and inferior parietal cortex. Similarly, spontaneous presence hallucinations in neurological patients have been associated with the temporoparietal cortex[27] and frontoparietal cortex[28,64]. Interestingly, temporoparietal cortex[65], and frontoparietal cortex[66,67] are key areas of bodily self-consciousness and are areas that integrate sensorimotor or multisensory bodily signals, as shown in human[68] and non-human primates[69,70]. Neural recordings in monkeys showed that the neural activity of bimodal visuo-proprioceptive neurons located in these regions were modulated when seeing body parts, but not when seeing objects[69,71–73]. Similar selectivity has been observed in humans[74,75]. Numerosity estimation has been associated with similar brain regions (especially with the intraparietal sulcus)[76–78]. Based on these findings we suggest that the selective overestimation in the human numerosity estimation task, but not for objects (in both healthy controls and PD patients) is due to the recruitment of human body-specific activations overlapping with those of human numerosity estimation. Future imaging work, using numerosity estimation and robotically induced presence hallucinations, should investigate this and validate these hypotheses.

Concerning virtual, augmented and mixed reality in medicine, immersive VR has recently emerged as a prominent tool for supportive treatment in mental health disorders[79] and rehabilitation[80], with recognized methodologies for validating its use for clinical interventions[81]. The use of VR as a diagnostic tool, however, remains relatively underexplored[82], despite promising studies in mental health and cognitive function[83,84]. Moreover, the present procedure in experiment 1 is based on the integration of immersive VR and robotics, allowing us to go beyond previous findings[22,28,50–52], by describing classical numerosity estimation effects for complex scenes with 3D virtual human agents and 3D objects[39,40,43] and by revealing their dependence on presence hallucinations. We also show that immersion in VR allows the reproduction of the same exposure and experimental conditions, for all participants, as immersive VR allows to provide the task instructions in a pre-recorded yet interactive manner, thereby automatizing the experimental procedure (i.e., the 3D virtual experimenter provides the experimental instructions in VR, physical interactions with the real experimenter are minimized) and limiting potential experimenter biases[82,85–87]. Moreover, because the exact same VR program can be executed by other researchers and in other settings, the experiment can be reproduced with minimal differences, thus enhancing the comparability between studies by different research groups and improving double-blind experimental designs and multi-center studies[85].

There are several limitations of our study. First, in study 1, although the overestimation bias was found in the human numerosity estimation task, but not the object numerosity estimation task, we only compared the effects of 0 ms (synchronous) and 500 ms delay (asynchronous) sensorimotor conflicts on the presence hallucinations. As robot-induced presence hallucinations have been shown to depend on the degree of sensorimotor conflict[22], future studies should test multiple delays of sensorimotor conflict[22], further defining the link between the intensity of robot-induced presence hallucinations and the overestimation of virtual human agents. Second, in study 1, participants are immersed in a virtual environment where a virtual character speaks to them (the virtual experimenter) and where several others enter and leave the room where the participant feels present. Our manipulation leverages on the subjective experience of copresence (togetherness with others in the virtual world[88]) that occurs in such VR simulations. Although copresence is often observed in VR with similar settings and rendering quality, our study could benefit from a direct assessment of copresence. Third, the data of Study 2 were acquired through an anonymized home-based online measurement, with limited information regarding the patients' symptoms and other clinical variables. Future clinical work should explore the novel digital presence hallucination marker jointly with detailed neurological, neuropsychological

and other clinical evaluations. Fourth, we carried out a cross-sectional study. Future work may also perform longitudinal assessments investigating for example the stability of the human numerosity estimation task marker over time, whether it reflects hallucination frequency, and how it relates to other hallucinations. Fifth, although we instructed PD patients in study 2 to perform the task while they were alone in the room, due to the online nature of study 2, we cannot ascertain that participants were actually alone during the experiment. The potential presence of other persons may have an impact on human numerosity estimation task and requires further investigation. Sixth, while we tested our online human and object numerosity estimation tasks in PD patients, future work should explore whether healthy controls experiencing spontaneous presence hallucinations also exhibit a human overestimation bias as compared to healthy controls (not experiencing such hallucinations). Seventh, although we specifically designed the control condition (object numerosity estimation task) with objects that are not close in nature to a hallucinated presence, future studies should design additional control conditions and test numerosity estimation for several other control objects, ranging from scrambled virtual humans to inverted virtual humans to living non-human animals and yet other control objects. These studies will be important for the cognitive neuroscience of visual human perception and how this is potentially modulated by an experimentally induced invisible hallucinated person. Eighth, study 1 used 3D VR to present stimuli and study 2 used 2D screens. As 3D VR (study 1) is a more ecologically valid approach (for testing the perception of humans) as compared to a 2D representation (study 2), future work should test PD patients in VR, which may potentially lead to stronger effects.

In conclusion, the present data demonstrate that the VR human numerosity estimation task and the online human numerosity estimation task are new digital markers for presence hallucinations, in healthy individuals as well as in patients with PD. By merging robotics, VR technology, and numerosity estimation, we report that experimentally induced presence hallucinations in healthy participants results in human numerosity overestimation (but not in the control object numerosity estimation task), revealing that human numerosity estimation task is a quantitative and robust digital marker for an experimentally induced hallucinatory mental state. By translating this VR-based marker to a home-based online assessment, in patients with PD, we show that online human numerosity overestimation is a digital marker for disease-related presence hallucinations that PD patients experience in their daily lives. Using a combination of controls, we ruled out the possibility that these effects stem from clinical variables or task demand. The present digital marker advances previous hallucination evaluations, as based on verbal interviews, by proposing a robust and quantitative assessment of these subjective mental states that are highly prevalent in many neurological and psychiatric diseases[8]. Our online home-based procedure strongly improves accessibility that often limits the impact of new laboratory-based markers and allows to reach people living far away from medical centers without requiring any specific equipment or trained staff[55,58]. As there is growing evidence that specific hallucinations, such as presence hallucinations, are an early marker for cognitive decline and dementia in PD[8,9,21,22], the human numerosity estimation task may not only detect proneness for psychosis, but also for later cognitive decline, requiring further longitudinal work.

## Methods
### Experiencing an invisible presence increases visual overestimation bias for numerosity estimations of visual virtual human agents (study 1)
**Preregistration.** The study preregistration is available at https://doi.org/10.17605/OSF.IO/YR3CP[49].

### Study population
Twenty-height healthy participants (18 women, 10 men; age ranging from 18 to 33 years, mean ± SD age = 24 ± 3.42 years) took part in this experiment. They were all right-handed according to the Edinburgh Handedness Inventory[89] (score ranging from 50 to 100, mean ± SD score = 87.5 ± 18). All the participants signed a written informed consent before participating in the experiment and were rewarded for their time with monetary compensation (CHF20/hour). None of the participants had current nor history of neurological, psychiatric and substance abuse disorders. Participants were also screened for a good stereoscopic vision through a stereoscopic acuity test[90]. At the end of the experiment, participants VR related experiences were assessed with a questionnaire about typical symptoms and effects induced by the experiment and virtual reality (Supplementary Note 8). All participants included in the study were naive to the purpose of the experiment. The experimental procedures (under protocol reference n° 2015-00092) were approved by the Cantonal Ethics Committee of Geneva (Commission Cantonale d'Ethique de la Recherche sur l'Être Humain - CCER, Switzerland).

### Apparatus and materials
**Robotic system.** The robotic system is composed of a commercial haptic interface (Phantom Omni, SensAble Technologies), coupled with a custom three degree-of-freedom robot in the back[91]. Participants are sitting on a chair and controlling the front robot situated on a table directly in front of them with their right index finger. The back robot is located behind their back and reproduces the movements initiated with the front robot with virtually no delay in the synchronous condition, and with 500 msec delay in the asynchronous condition, which has been shown to induce presence hallucinations. This creates different degrees of sensorimotor conflict between the right-hand movement and the somatosensory feedback on the back[28].

**Virtual reality system.** The virtual reality system consists in a commercial head-mounted display system (Oculus Rift CV1 coupled with a single Oculus Rift Sensor).

**Auditory system.** The auditory system consists of a commercial Auditory headset (Arctis Wireless Pro).

**Graphic system.** The graphic system consists in a laptop computer running Windows 10 and equipped with an Intel i7 6700HQ processor, 16 Gb Random-Access Memory, and a Nvidia GeForce GTX 1060 graphic card, ensuring a constant 90 Hz display rate of the experiment in the virtual reality system. The virtual reality system is plugged into the graphic system. The experiment running on the graphic system is implemented in Unity 2019.3.13f1 using C#.

**Experiment control and progress tracking system.** The experiment is controlled, and progress tracked in real time through a mobile-compatible application, implemented in Unity 2019.3.13f1 using C#. The device used consists of a smartphone running Android (Samsung Galaxy S8+), connected to the graphic system over Wi-Fi.

### Virtual experimenter
The virtual experimenter was created with Character Creator and the Headshot plugin. The model of the virtual experimenter is available as supplementary material (see Code availability section). All the instructions are given by the virtual experimenter (Supplementary Movie 5; Supplementary Movie 6; Supplementary Movie 7), whose lip movements are automatically synchronized with the prerecorded audio using speech-driven lip-sync[92].

## Numerosity stimuli

Visual stimuli were generated on Unity 3D (version 2019.3.13f1). The virtual environment was modeled in 3DS max and consisted of a realistic representation of our experimental room. The model of the virtual environment is available as supplementary material (see Code availability section). The virtual scene exposure was set to high in order to have a low global brightness, to encourage hallucinations induction which have been reported to often be triggered in low levels of illumination in patients' daily life[93–95]. Stimuli consisted of a 3D scene of our virtual experiment room with virtual human agents (or control objects) inside. Virtual human agents and control objects were placed in the virtual scene in front of the camera viewpoint, facing the viewpoint in a range from – 90° to 90°, and in a way that they do not overlap completely from the viewpoint. Virtual human agents and control objects were placed between 1.75 m and 5.25 m in depth from viewpoint, and between −1.5 m and 1.5 m from right to left. Array of virtual human agents and control objects occupied a maximum virtual camera visual angle of 60° horizontally. This ensures that virtual human agents and control objects were located within the participants' peripheral field of view, which is the "vision produced by light falling on areas of the retina outside the macula"[96], and that they were displayed inside and close to the limit of 30° retinal eccentricity, which is the limit from which the visual acuity decreases more strongly[97]. The virtual human agents and control objects arrays were ranging from 5 to 8 and generated on the fly based on seeded configurations. The lower bound was decided as the lower bound of the estimation range of our type of stimuli (Supplementary Note 1).

## Robot-induced subjective experiences questionnaire

Participants performed the robot-induced presence hallucination paradigm for 2 minutes, one time in synchronous condition and one time in asynchronous condition, randomized order across participants. After each condition, a lab-tailored questionnaire was used to measure the presence hallucination ("I felt as if someone was standing close to me (next to me or behind me)"), along with other subjective experiences as self-touch ("I felt as if I was touching my back myself"), passivity experience ("I felt as if someone else's was touching my back"), and loss of agency ("I felt as if I was not controlling my movements or actions"). Three control questions were also asked ("I felt as if someone was standing in front of me", "I felt as if I had two body", "I felt anxious/stressed"). These 7 questions measuring presence hallucinations and other illusions were adapted from previous work with the sensorimotor robotic device[28]. At the beginning of the robot-induced presence hallucination paradigm, participants were put in a dark environment, and following an acoustic cue (beep, 1000 Hz, 250 msec) they were asked to close their eyes and start performing the poking movements with the front-robot. During the whole trial duration white noise was presented through headphones to isolate participants from the robotic noise. After 120 sec a second acoustic cue (double beep, 1000 Hz, 250msec each beep separated by 250 msec) indicated the end of the trial, and participants were asked to answer questions displayed in the head-mounted display. The questions were displayed in a randomized order across conditions (synchronous and asynchronous) and participants. Participants were asked to indicate on a 7-point Likert scale how strongly they felt the sensation described by each item (from 0 = not at all, to 6 = very strong). The yaw orientation of the head-mounted display was used to target a value on a slider. Once the desired value was targeted, participants were instructed to say verbally either "ok" or "validate" to validate their current answer. Validation was performed by voice recognition (Windows Speech KeywordRecognizer). An overview of the robot-induced subjective experiences questionnaire protocol is shown in Supplementary Fig. 1. A complete overview of the virtual instructions is available in Supplementary Movie 5.

## Numerosity estimation

The numerosity estimation task was divided into two parts, one presenting arrays of virtual human agents (human numerosity estimation task) in the virtual environment and the other one presenting arrays of control objects (object numerosity estimation task) in the virtual environment, randomized order across participants. Each part contained four blocks of respectively twelve, eight, eight and twelve trials. Blocks of twelve (respectively height) trials contained three (respectively two) times each numerosity. Each block started with a 60 sec habituation phase, during which several virtual human agents moved and discussed in the virtual environment in the virtual human agents part (around 10 in total per habituation phase) (Supplementary Movies 3, 4). Virtual human agents could navigate in the whole virtual room, sometimes passing on the side or behind the participant. Participants were implicitly invited to look around and observe the whole virtual environment during these habituation phases. Some of them (between 2 and 3 depending on the habituation phase) were discussing together. There were a total of four different habituation phases, presented in the same order across participants. In the control (object) condition, objects (boxes with a wireframe shader) materialize and dematerialize over time on the path (position and orientation matched with those of the virtual human agents) in the corresponding habituation phase (Supplementary Movie 4 shows a direct comparison of virtual human agents and objects habituation phases). During the virtual human agent's habituation phases, the virtual human agents perform a succession of pre-recorded and pre-determined motion-captured animations (animations are looped and posed-matched). During the virtual human agent's habituation phases, audio footsteps are spatially rendered at the location and timing of each virtual human agent's foot new contact with the ground. During the corresponding control object habituation phases, a new control object materializes each 3 animations of each virtual human agent at its position and with its orientation. This control object then dematerializes after 5 animations of the corresponding virtual human agent. In the control objects condition, control objects replacing, and matching position and orientations of virtual human agents appeared and disappeared along time in the virtual environment (Supplementary Movie 4). At the end of a habituation phase, trials started. At the beginning of a trial, participants were put in a dark environment, and following an acoustic cue (beep, 1000 Hz, 250 msec) they were asked to close their eyes and start performing the poking movements with the front-robot. The back robot sensorimotor stimulation was either synchronous (0ms delay) or asynchronous (500 ms delay), alternating across blocks, starting condition equally balanced across participants. During the whole trial duration white noise was presented through headphones to isolate the participant from the robotic noise. After 30 sec a second acoustic cue (double beep, 1000 Hz, 250msec each beep separated by 250msec) indicated the end of the trial. Participants then had to open their eyes and keep their gaze on a central fixation cross, which was briefly presented (range between 500msec and 1500 msec) before the presentation of the visual stimulus. The visual stimulus, consisting in an array of virtual human agents or control objects (ranging from 5 to 8) in the virtual environment, was then presented in front of the participant for a duration of 200 msec. Participants then had to report the number of virtual human agents ("How many people are in the room?") or control objects ("How many objects are in the room?") they estimated to be in the virtual environment on a scale ranging from 0 to 20. As in task 1, the yaw orientation of the head-mounted display was used to target a value, and voice recognition (Windows Speech KeywordRecognizer) was used to validate the answer. There was a total of 10 different stimuli configurations possible per numerosity (ranging from 5 to 8), where stimuli type (virtual human agents or control objects) had a specific configuration (in terms of stimuli position and orientation, which were matched between virtual human agents and control objects for each configuration). Half of the participants were

associated with 5 stimuli configurations per numerosity, the other half of the participants were associated with the 5 remaining stimuli configurations per numerosity. An overview of the numerosity estimation protocol is shown in Supplementary Fig. 1.

### Data analysis

**Robot-induced sensations questionnaire.** Cumulative link mixed model (packages ordinal[98] and RVAideMemoire[99] in R[100]) were used to analyze questions of the robot-induced sensations questionnaire. Sensorimotor stimulation (synchronous or asynchronous) was set as a fixed effect, and random intercepts for each subject were assumed. The significance of fixed effects was estimated with likelihood Ratio Test comparing full model (with sensorimotor stimulation set as fixed effect) against a reduced model without the fixed effect in question. We concluded that the fixed effect (condition) was significant if the difference between the likelihood of the two models was significant.

**Numerosity estimation.** Linear mixed effects models (packages lme4[101] and lmerTest[102]) with robotic sensorimotor stimulation (synchronous or asynchronous), presented numerosity and type of stimuli (human and control objects) as fixed effect, random intercept for each subject was performed on the numerosity estimation data. The significance of fixed effects was estimated with likelihood Ratio Test. Post-hoc analysis was performed on the significant interactions and corresponded in pairwise comparisons using independent-samples t-tests, not corrected for multiple comparisons.

The general estimation performance at each numerosity was assessed with one-sample t-tests against presented numerosity, with reported p-values not corrected for multiple comparisons. The difference of estimation between human and object numerosity estimation tasks was assessed with paired sample t-tests at each of the presented numerosity, with reported p-values not corrected for multiple comparisons.

Trials in which participants did not see the stimuli (answer 0 or 1) were excluded. This procedure resulted in the exclusion of no trials across conditions and participants.

**Mediation analysis.** Linear mixed effects models with robotic sensorimotor stimulation (synchronous or asynchronous) as fixed effect, random intercept for each subject was performed on the human numerosity estimation data to estimate the effect of the robotic sensorimotor stimulation on human numerosity estimation. Linear mixed effects models with robotic sensorimotor stimulation (synchronous or asynchronous) as fixed effect, random intercept for each subject was performed on the robot-induced sensation questionnaire data to estimate the effect of the robotic sensorimotor stimulation on robot-induced presence hallucination question rating. Linear mixed effects models with robotic sensorimotor stimulation (synchronous or asynchronous), robot-induced sensation questionnaire data (robot-induced presence hallucination question rating) as fixed effect, random intercept for each subject was performed on the numerosity estimation data to estimate the effect of the robot-induced presence hallucination question rating on human numerosity estimation, controlling for robotic sensorimotor stimulation. The significance of fixed effects was estimated with likelihood Ratio Test. Causal mediation analysis (package mediation[103]) was performed on the previous models to estimate the indirect effect of robotic sensorimotor stimulation (synchronous or asynchronous) on human numerosity estimation by robot-induced sensation questionnaire data (robot-induced presence hallucination question rating). The significance of the indirect effect was tested using 1000 bootstrapped samples, and the 95% confidence interval was computed by determining the indirect effects at the 2.5th and 97.5th percentiles.

### Online numerosity estimations of visual virtual human agents reveals presence hallucinations in Parkinson's disease (study 2)

In this online web-based experiment, participants first filled in some socio-demographic information, answered a questionnaire on alteration of perception corresponding to the frequency of hallucinations occurrence in daily life, followed by a screen calibration procedure, and the numerosity task. The experiment was available in French and English. The experiment could be performed on a computer or on a tablet. Participants were instructed to perform the experiment while being alone in the room.

### Study population

One hundred and seventy patients with PD (93 women, 77 men; age ranging from 42 to 79 years, mean ± SD age = 65.4 ± 7.83 years; PD duration ranging from 1 month to 25.6 years, mean ± SD PD duration = 6.44 ± 5.19 years) took part in this study. Of the 170 patients with PD, 118 patients of interest to answer our research question were kept for the analysis of the human and object numerosity estimation tasks in the current paper (see Data analyses section below for details). All participants consented to voluntarily participate in the study, prior to the beginning of the experiment. This study was considered as falling outside of the scope of the swiss legislation regulating research on human subjects, so that the need for local ethics committee approval was waived (Commission Cantonale d'Ethique de la Recherche sur l'Être Humain – CCER, Switzerland – Req-2021-00378).

### Socio-demographic information

At the beginning of the experiment, participants were asked to indicate their gender, age, country, time, visual disturbances, and whether they had been diagnosed with Parkinson's disease. In case of positive answer to this last question, participants were also asked the date of diagnosis (year and month), the side of the body where symptoms appeared first ("left", "right", "both", "I don't know"), the current medication along with the daily dosage, and the time of the last medication intake for Parkinson's disease (Levodopa).

### Questionnaire on alteration of perception

This questionnaire is a self-assessment questionnaire on whether the frequency of occurrence of specific hallucinations in daily life (passage hallucination, presence hallucination, visual illusion, or visual hallucination – see Supplementary Table 6 for the list of questions). Participants answered on a 5-item Likert scale (0 – Never; 1 – Rarely (less than once a month); 2 – Occasionally (several times, but less than once a week); 3 – Frequently (several times a week, but less than once a day); 4 – Daily (almost every day, several times a day); see Supplementary Table 8 for the report of these occurrences).

### Screen calibration

Participants were invited to measure and report the length of a line displayed on their screen. This measure allows scaling the stimuli so that their physical size on the display is the same for all participants, independently from the physical size of the monitor screen or tablet used by the participant. In addition, in order to approximate a controlled viewing angle for all participants, we adopted a stimuli presentation ratio of 2:1 based on the device used (stimuli displayed on computers were two times bigger than those displayed on tablet), in combination with a viewing distance ratio of 2:1. That is, participants were instructed to stay one meter away from their screen on a computer and fifty centimeters away from their screen on a tablet. These values were chosen based on the US Occupational Safety and Health Administration (OSHA) ergonomics research viewing distance recommendations (from the eye to the front surface of the computer screen, between 20 and 40 inches, i.e. 50 and 100 cm (https://www.osha.gov/etools/computer-workstations/components/monitors)).

The detection of whether the participants were using a computer, or a tablet was automatic.

## Numerosity task

The numerosity estimation task was divided into two parts, one online human numerosity estimation task and one online object numerosity estimation task, randomized order across participants. Each part contained 40 trials, 10 trials per numerosity (ranging from 5 to 8), randomized order across participants. There was a total of 10 different stimuli configuration possible per numerosity (ranging from 5 to 8), which were the same as in the behavioral experiment (numerosity estimation as an implicit measure for robot-induced presence hallucinations). The stimuli were 1280px width by 720px height pictures and measured onscreen approximately 35.2 ×19.8 cm on the computer version and 17.6 × 9.9 on the tablet version. The stimuli were displayed for 250ms[38,39,104]. The increase in stimulus presentation duration from study 1 (200 ms) to study 2 (250 ms) was based on pilot data in elderly healthy participants, adapting the difficulty of our human and object numerosity estimation tasks from young healthy participants (Study 1) to patients with Parkinson's disease (Study 2).

## Apparatus and material

This online experiment was developed in house using javascript and the jsPsych library[105] (on client side: javascript, html, css; on server side: https server in node.js, nginx as a reverse proxy, running in two docker containers) and was hosted on an EPFL server in a dedicated Virtual Machine (1xvCPU, 1GB RAM, 40GB HDD) in demilitarized zone. Data was saved and stored on EPFL's servers.

## Data analysis

**Numerosity estimation.** Of the 170 patients who participated in our online study, we included a total of 118 PD patients in the analysis of the numerosity estimation task: 63 PD patients with presence hallucinations (PD-PH) and 55 PD patients without any hallucinations (PD-nH). The selection criteria are described below. Participants who had a very low refresh rate (less than 20Hertz) or resolution (less than 800px on both axis) were excluded from the analysis of the numerosity task. This resulted in the exclusion of respectively 4 and 4 participants. Participants who reported visual disturbances that could negatively impact the task were also excluded from the analysis of the numerosity task (mainly diplopia, the reports of visual disturbances of these participants can be found in Supplementary Table 7). This resulted in the exclusion of 10 participants. Participants who reported hallucinations, but not presence hallucinations (passage hallucinations, visual illusions or structured visual hallucinations, $n = 39$) were also excluded from the current analysis. In the selection procedure described above, some participants belong to different categories of rejection criteria.

Trials in which participants did not see the stimuli or made an evident mistake in their reporting (answer less or equal to 3 or answer superior to 50), along with trials in which participants took too much time to give an answer (response time superior to 15 seconds) were excluded from the analysis of the numerosity task. This procedure resulted in the exclusion of 2.2% of the trials in the human numerosity estimation task (104 trials over 4720) and 2.5% of the trials in the object numerosity estimation task (120 trials over 4720).

Participants were then separated into two groups based on their response to the questionnaire on alteration of perception (Supplementary Table 6). The first group (PD-nH group) contains PD patients not having experienced any kind of hallucination (response equal to 0 to all questions). The second group (PD-PH group) contains PD patients having experienced presence hallucinations (response greater than or equal to 1 to the corresponding question). The demographic and clinical characteristics of the PD-nH and PD-PH group population are reported in Table 1. The technical specifications of the devices used

by the PD-nH and PD-PH group population are reported in Supplementary Note 9.

Linear mixed effects models with PD group (PD-nH or PD-PH), presented numerosity and type of stimuli (virtual human agents and control objects) as fixed effect, random intercept for each subject was performed on the numerosity estimation data. The significance of fixed effects was estimated with likelihood Ratio Test. Post-hoc analysis was performed on the significant interactions and corresponded in pairwise comparisons using independent-samples t-tests, with reported p-values not corrected for multiple comparisons.

The general estimation performance at each numerosity was assessed with one-sample t-tests against presented numerosity, with reported p-values not corrected for multiple comparisons.

The difference of estimation between human and object numerosity estimation tasks was assessed with paired sample t-tests at each of the presented numerosity, with reported p-values not corrected for multiple comparisons.

**Clinical variables.** In Table 1, gender independence was compared between groups (PD-nH vs PD-PH) with Pearson's chi-squared test with Yates' continuity correction. Age, PD duration and Levodopa equivalent daily dose (LEDD) were compared between groups (PD-nH vs PD-PH) with two-sample t-tests. LEDD was calculated as a sum of the conversion of each parkinsonian medication to Levodopa Equivalent Dose[106–108]. Four PD patients did not correctly report their medication (two PD-nH and two PD-PH); thus, they were excluded from the calculations and statistical test of LEDD.

### Reporting summary

Further information on research design is available in the Nature Portfolio Reporting Summary linked to this article.

## Data availability

The main data supporting the results in this study are available within the paper and its supplementary Information. The source data have been deposited at https://doi.org/10.5281/zenodo.10511579. Source data used to generate figures are provided with this paper as a Source Data file. Source data are provided with this paper.

## Code availability

All codes, executables, scripts, models, stimuli to reproduce the findings have been deposited at https://doi.org/10.5281/zenodo.10511579.

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

## Acknowledgements

This research was supported by the Swiss National Science Foundation (grant number SNF 320030_188798) to O.B.; two generous donors advised by CARIGEST SA (Fondazione Teofilo Rossi di Montelera e di Premuda and a second one wishing to remain anonymous) to O.B.; Bertarelli Foundation to O.B.; Empiris Foundation to O.B. The funders played no role in the study design, data collection, analysis and interpretation of data, or the writing of this manuscript. The authors thank all patients for their participation in study 2, as well as Parkinson Schweiz, Parkinson's UK and Association France Parkinson for their help in recruiting patients with PD for study 2. The authors thank all participants for their participation in Study 1. The authors thank Mr. Laurent Jenni for his robotics support. The authors thank Mr. Arthur Trivier for his help in the design of the robotic system figure part used in Fig. 1 and his help in the design of the desk figure part used in Fig. 5.

## Author contributions

L.A., F.B., B.H. and O.B. designed study 1. L.A., J.P., F.B. and O.B. designed study 2. L.A. realized the technological developments of study 1 and study 2. F.B., B.H., O.B. supervised study 1. F.B., J.P., O.B. supervised study 2. L.A. recruited participants and performed the experiments in Study 1. L.A. analyzed the data in Study 1 and Study 2. L.A. and J.P. recruited patients in study 2. J.P. coordinated patients recruitment via patient associations: Parkinson Schweiz, Parkinson's UK and Association France Parkinson. J.P. designed the questionnaire for the assessment of hallucinations in Study 2. L.A., F.B. and O.B. wrote the manuscript. All authors provided critical revisions and approved the final version of the manuscript.

## Competing interests

A patent application has been submitted (application number EP23154061.8 – Numerosity estimation impairment measurement system), describing a solution for measuring and quantifying an impairment in numerosity estimation in human subjects, based on the methods described in this manuscript, with Ecole Polytechnique Fédérale de Lausanne (EPFL) as patent applicant, and L.A., O.B., F.B., B.H., and J.P. as inventors. O.B. is inventor on patent US 10,286,555 B2 held by the Swiss Federal Institute (EPFL) that covers the robot-controlled induction of presence hallucination. O.B. is inventor on patent US 10,349,899 B2 held by the Swiss Federal Institute (EPFL) that covers a robotic system for the prediction of hallucinations for diagnostic and therapeutic purposes. O.B. is co-founder and shareholder of Metaphysiks Engineering SA, a company that develops immersive technologies, including applications of the robotic induction of presence hallucinations that are not related to the diagnosis, prognosis or treatment of Parkinson's disease. O.B. is a member of the board and shareholder of Mindmaze SA. The authors declare no other competing interests.
