## [Peer Review File · Nature Communications]

Numerosity estimation of virtual humans as a digital-robotic marker for hallucinations in Parkinson's diseaseEditorial Note: Parts of this Peer Review File have been redacted as indicated to remove third-party material where no permission to publish could be obtained.

REVIEWER COMMENTS

Reviewer #1 (Remarks to the Author):

I think this is an interesting work, suitable for high-profile publication after revision for clarity. As I understand, hallucinations are prevalent in Parkinson's, and could be a useful early marker. This study uses an interesting approach, where they ask participants to estimate the number of human figures, and use overestimation evidence for hallucination (presumably the hallucinated figures are confused with the images). It also uses many clever and interesting techniques designed by the authors to induce hallucinations in typical participants.

Although the English is near-perfect and the writing generally good, I found it very hard to appreciate what they had done. One reason is the excessive use of unorthodox and unnecessary acronyms. We can accept VR, but NE, rather than estimated number? Even on the graph, with presented numerosity as abscissa? Then we go to NEH (estimation of humans) and NEO. Even hallucinations, the main goal of the study, are disguised as PH. This is problematic, as the definitions are given only once, and hard to find (not even in the figure captions), making it difficult to scan the paper. Even the sub-titles are dominated by acronyms. More importantly, they give the manuscript an air of mystique, suggesting that something very exotic is being measured (NEH), rather than hallucinating extra figures, which anyone can instantly understand.

Similarly, the term "digital humans" is confusing, partly because in numerosity studies digits usually mean symbolic numbers, partly because digital (as opposed to analogue) usually means discrete, quantized. I do not know in what sense they are digital, other than they are computer-generated, and computers are digital devices. Call them human images or similar.

More importantly, but along the same lines, I would have appreciated some sort of theoretical explanation of what is going on. I understand the main issue of this manuscript is translational, and there are probably other more theoretical papers in the pipeline, but some theory would aid understanding. Where do the hallucinations come from? – some variant of the predictive coding ideas out there? Is it the case that the hallucinations blend with the human figures, leading to over-estimation? I presume something like this must be happening, but some discussion, without pre-empting future publications, would be appreciated here.

Finally, please put equality lines on the graphs, to make the over-estimation more obvious (there are symbols, but these get confused with data).

All that said, I think the work is great! With very minimal revision, it should make a fine contribution to Nature Communications.

Reviewer #2 (Remarks to the Author):

The paper by Albert et al. reports a study on 28 healthy controls and 118 PD patients, focused on Presence Hallucinations (PH), and on the possible link between PH and Numerosity Estimations of Human (NEH) figures in a Virtual Environment.

The statistical power of the study is adequate, the applied statistical corrections are state of art, the paper is a further evolution of several papers by the same authors, all published in highly respected journals, where also fMRI evaluations were obtained.

Shortly the study shows that :

- 1) In a Virtual Environment (eye goggles), when "Healthy Participants" are operating a virtual robot, and are involved in NE task, NEH errors (with increments as compared to actual stimuli) are reported. The NE do not increase if stimuli are not human figures.
- 2) These errors are furtherly increased if a distractor is employed : a mechanical arm poking on the back of the "healthy participant" after a 0.5 s delay from visual stimuli, poking has no effect if it is simultaneous with visual stimuli, it has no effect also if stimuli are not Human Figures.
- 3) In PD patients the same incremental NEH can be detected in a simpler Virtual Environment, i.e. the patient looking at the task from his home based computer or tablet, without any robotic task or distracting poking.
- 4) These NEH seem to correlate to PH ,as reported subjectively, in "Healthy participants" and PD patients.

Extracting this information from the paper is difficult, the presentation is dispersed in repetitive statements, the consistency, in terms of number of participants, is not immediately underlined, it is difficult to understand that Study 1 was only done in controls and Study 2 only in PD patients. The authors make reference to their previous studies in PD patients, in order to support the logic of the study presentation, but my feeling is that supporting with prior studies, in this case, is not methodologically acceptable.

While I cannot challenge the appropriateness of methods of data collection and statistical evaluation, I found that the design of the study and its conceptualization has some flaws.

Design :

- 1) Why the set of experiments is not the same for the different groups (Healthy Controls-HC, PD with PH, PD without PH)? We do not know if HC are prone to PH also in a familiar environment, or it is only in the secluded virtual reality of eyegoggles that PH occurs.
- 2) Why is not poking performed also in the home based experiment? Knowing about the effect of distractors in familiar or unfamiliar environments would add relevant information.

Conceptualization:

Presence Hallucinations (PH) are a small aspect of hallucinations in PD; with passage hallucinations and illusions, PH are considered the typical triad of "early" hallucinations, i.e. occurring early in the course of PD, as can be found in papers quoted in the present study. Several descriptions show that early hallucinations can wane, to reappear later in advanced PD, shortly followed by complex hallucinations. One of the studies quoted in the present paper (Bejir-Kasem et al.) shows that a Default Mode Network disruption can be observed in early hallucinators, and it is similar to what observed in late - hallucinators, but the clinical aspect and the insight are extremely different, the amount of distress makes

the difference, and the relevant therapeutic target is likely to be the distress linked to complex hallucinations, as it is well known that the early hallucination wanes when the PD patient focuses his attention to it.

Therefore the first question would be: given the clinical relevance, why was PH considered an important target?

Methods to induce increments of hallucinations were already described, i.e. Pareidolia tests for Illusions (Uchiyama et al. Brain 2012) and Virtual Environment for Early and Late Hallucinators (Onofrj et al. J Neurol Sci 2006). We know, from these studies, that a virtual environment increases hallucinatory percepts in PD, in early and late hallucinators.

Thus the second question is: why was not the effect of Virtual Reality compared in other tasks?

Finally, in their introductory claim, the authors state that assessment of hallucinations is biased by verbal report, as obvious, but the correlation of NEH with PH is again only based on a verbal, subjective, report by the patient, or control, who says he/she had PH.

To me, this is the major logical challenge. More solid correlation of NEH and hallucinations could be obtained, in example, by selecting patients on the basis of fMRI alterations.

Minor:

line 432- "elderly PD patients" emerge from nowhere in the study. When was an age stratification performed?

Line 174 and 348 – "subitizing" is an unusual English word, mostly used in psychology studies, the authors should explain it, and should also say why they believe that subitizing for numbers above 5 is acceptable.

Suggested readings:

Collerton D et al. Neurosci Biobehav Rev, 2023

O'Brien D et al. J Neurol Neurosurg Psychiatry, 2020

Onofrj et al. Mov Dis, 2015 and 2019

Reviewer #3 (Remarks to the Author):

Review for "Digital-robotic markers for hallucinations in Parkinson's disease" in Nature Communications

Summary: in this article, the authors present two studies. The first demonstrates in young, healthy patients the usefulness of a robotic protocol combined with virtual reality (VR) to i) induce presence hallucinations (PH) and ii) study the association of the latter with numerosity estimation (NE) of virtual human agents. The second presents a study carried out remotely by Parkinson's patients, with or without PH, and attempts to show the association between the latter and the NE of virtual human beings. Together, these two studies attempt to demonstrate the efficacy of NE as an objective and quantitative marker of PH, while presenting a remote protocol supported by laboratory results.

Disclaimer: as I'm not an expert in Parkinson's disease, I can't assess the theoretical relevance of evaluating and quantifying PH. I will focus on my field, which corresponds to the virtual experimental protocol, notably on the methodological aspects of VR and the extrapolation of results, as well as on the sense of (co)presence.

General criticism: both the experimental and analytical methodology used in these two studies appear to be rigorous. The demonstrative reasoning is clear and follows logically from the results presented by the two studies, although I find that the presentation format would benefit from being a bit more orthodox. If fellow reviewers who are experts in the theoretical and clinical field endorse the paper's publication by raising its relevance, I certainly won't object. However, I think the paper could gain in rigor by clarifying the points I am now describing. Indeed, I find that there are points of importance that are not discussed in the article and should be:

- 1. The NE stimulus is sometimes indicated as lasting 200ms and sometimes 250ms. I don't think this difference is justified in the manuscript. Please clarify whether this is a drafting error or whether there really is a difference, and if so to justify it.

- 1. I fully understand the interest in using VR in this context, and I think it could quite possibly make the task more discriminating and relevant, by appealing to ecological cognitive mechanisms for detecting people (e.g. 3D movements, posture...) that are much harder to trigger in 2D. However, reading the article and the reasoning behind it suggests a logical continuation, a transposition of Study 1, in VR, to Study 2, on computer/tablet. It seems to me that these two media involve different cognitive and motor perceptual processes. However, these differences are not discussed in the article when articulating Study 1 and Study 2. I believe that a discussion on this subject could greatly enrich the manuscript. For example, in Study 1, NE was reported using a gauge to assess the ordinal distance between numbers, whereas in Study 2 it was an open-ended question with no visual indication. I don't have any empirical references of an effect on this specific subject, but I think studies somehow intended for comparison should try to follow a similar methodology on these kinds of aspects. I think this (and other aspects of comparison) deserve a point of discussion.

- 3. You compare the differences in reaction times (not significant) to say that "Additional analysis ensured that the observed differences between NEH and NEO are not explained by differences in task difficulty.". I think it's a bit bold to assert such a thing on the sole basis of reaction times between conditions, all the more so for a response based on a very short stimulus (200ms) that no longer exists at the moment of response. What's more, I'm not convinced that reaction time is sufficient to say that there is no difference in difficulty between the two tasks. I invite the authors to nuance their remarks here and add a quick discussion on the subject. For example, it seems to me that the bright green of the control condition could make the task easier, compared with the dark colors of the individuals. This could well moderate or modulate performance.

- 4. If I'm not mistaken, you give very little information on head movements in virtual reality. What were the instructions during habituation phases with humans or objects? Were participants free to move? Did they have to look straight ahead all the time? Was it natural to look straight ahead, as there was nothing

to look at on the sides or behind? This needs further clarification. Head rotations can have significant effects in virtual reality, particularly on cybersickness. I think it would be a good idea to add a word about cybersickness to the method. Did any participants complain?

- 5. When watching habituation videos with humans, some virtual agents come and stand behind the participant's angle of view, so that they disappear. With videos, it is very difficult to determine whether this effect also exists in virtual reality, which could lead to head movements (see previous point). However, depending on the instructions, if virtual human beings pass behind the subject and the latter has no opportunity to turn around, it is quite possible that this will have an impact on the perception of PH and co-presence (especially considering the tactile stimuli behind the participant's back). Please clarify this point

- 6. In addition to all this, I'm surprised by the choice of objects in the control condition and their movement animation. Why not choose something closer to human beings? Indeed, the objects: i) appear and disappear rather than move, which greatly complicates their overall apprehension, ii) don't seem to come, one by one, through the front door as humans do in some videos, iii) never seem to pass behind the participant as humans do, iv) never seem to stand still as humans sometimes do. All these points deserve important clarification.

- 7. I think the general behavior of individuals/objects during the habituation phases deserves more detail. How were their movements decided?

- 8. Some Post-hoc comparisons are carried out with p-value corrected for multiple comparisons, while others are not. I think this deserves a sentence of justification. In addition, I'm not sure that confidence intervals and effect sizes are systematically reported. Please take the time to check that each statistical test includes, where applicable, the confidence interval and effect size rather than just the p-value.

- 9. It seems to me that the authors do not report any details on the conditions under which the PD participants took part in Study 2. However, this type of study, particularly as it includes a strong dimension of perceived human presence, deserves a word on the conditions under which it was carried out. Did you ask the participants to be alone? Did you control their presence? It seems to me that the number of humans actually present in the experimental room can have a significant impact on the results, and that this merits a point of discussion.

- 10. You note that "Participants were instructed to stay one meter away from their screen on a computer and fifty centimeters away from their screen on a tablet". You should justify this.

- 11. It seems that authors use abbreviations (e.g. NE) before the first occurrence of the term. Please take the time to check that each abbreviation is explained.

- 12. Finally, and although biased by my research, I would amplify in the article's discussion the potential importance of the sense of presence in this task. I'll measure the latter using questionnaires not only of co-presence but also of spatial presence, and assess its impact on participants' performance on the NE task as well as on their feeling of PH

Reviewer #1 (Remarks to the Author):

I think this is an interesting work, suitable for high-profile publication after revision for clarity. As I understand, hallucinations are prevalent in Parkinson's, and could be a useful early marker. This study uses an interesting approach, where they ask participants to estimate the number of human figures, and use overestimation evidence for hallucination (presumably the hallucinated figures are confused with the images). It also uses many clever and interesting techniques designed by the authors to induce hallucinations in typical participants.

We thank Reviewer #1 for the positive appreciation of our manuscript and the different suggestions that we believe allowed us to improve the manuscript.

Although the English is near-perfect and the writing generally good, I found it very hard to appreciate what they had done. One reason is the excessive use of unorthodox and unnecessary acronyms. We can accept VR, but NE, rather than estimated number? Even on the graph, with presented numerosity as abscissa? Then we go to NEH (estimation of humans) and NEO. Even hallucinations, the main goal of the study, are disguised as PH. This is problematic, as the definitions are given only once, and hard to find (not even in the figure captions), making it difficult to scan the paper. Even the sub-titles are dominated by acronyms. More importantly, they give the manuscript an air of mystique, suggesting that something very exotic is being measured (NEH), rather than hallucinating extra figures, which anyone can instantly understand.

We thank the reviewer for the comment and acknowledge the concerns about the excessive use of acronyms in our manuscript. We have revised the manuscript accordingly and hope that the readability of the manuscript has significantly improved.

The following acronyms have been removed from the manuscript:

- "NE" is replaced by "numerosity estimation".
- "NEH" is replaced by "human numerosity estimation task".
- "NEO" is replaced by "object numerosity estimation task".
- "PH" is replaced by "presence hallucination".
- "HMD" is replaced by "head-mounted display".
- "vm" acronym (virtual meters) has been removed, as used only once.

The following acronyms have been kept in the revised manuscript:

- "PD" acronym ("Parkinson's Disease") is kept because it is well accepted.
- "VR" acronym ("virtual reality") is also kept for the same reason.

- “PD-PH” acronym (“Parkinson’s Disease patients with Presence Hallucination”) and “PD-nH” acronym (“Parkinson’s Disease patients with no Hallucination”) are kept but are always redefined/explained when used in figures/tables captions.

Accordingly, acronyms have been removed from figure legends, titles, and captions. Equality lines have been added to graphs to make the over-estimation easy to visualize.

Figure 2. Numerosity estimation (study 1). General numerosity estimation performance for each tested numerosity in the (a) the NEH human numerosity estimation task and (b) the NEO object numerosity estimation task (study 1). Each dot indicates the individual NEH numerosity estimation task mean (~~over the trials~~) estimate for the corresponding presented numerosity. The dots with the bar on the right sides indicate the in-between subject mean for each presented numerosity. Note the general overestimation bias in the NEH human numerosity estimation task and the NEO object numerosity estimation task. Error bar represents 95% confidence interval.

a

a

c

c

Figure 3. **NEH Human and object numerosity estimation tasks and NEO** as a function of sensorimotor stimulation (synchronous, asynchronous) (study 1). (a) Task performance is shown for each presented numerosity in the **human numerosity estimation NEH** task, **separately** for the asynchronous (**dark blue**) and synchronous (**light blue**) **condition sensorimotor stimulation**. Each linked pair of dots indicates the individual **human numerosity estimation task NEH** mean estimate for the corresponding numerosity in asynchronous **condition** (dark blue) and synchronous **condition** (light blue) **sensorimotor stimulation**. The dots with the bar on the left and right sides indicate the mixed-effects linear regression between asynchronous (dark blue) and synchronous (light blue) sensorimotor stimulation for each tested numerosity. (b) **Human numerosity estimation task NEH** (asynchronous (**dark blue**) versus synchronous (**light blue**) **condition sensorimotor stimulation**). Each linked pair of dots indicates the individual **human numerosity estimation task NEH** mean estimate in asynchronous **condition** (dark blue) and synchronous **condition** (light blue) **sensorimotor stimulation**. The dots with the bar on the left and right sides indicate the mixed-effects linear regression between asynchronous (dark blue) and synchronous (light blue) sensorimotor stimulation. (c) **Object numerosity estimation NEO** (asynchronous (**dark red**) versus synchronous (**light red**) **sensorimotor stimulation**). Each linked pair of dots indicates the individual **object numerosity estimation task NEO** mean estimate in asynchronous **condition** (dark red) and synchronous **condition** (light red) **sensorimotor stimulation**. The dots with the bar on the

left and right sides indicate the mixed-effects linear regression between asynchronous (dark red) and synchronous (light red) sensorimotor stimulation. Error bar represents 95% confidence interval. **P ≤ 0.01. N.S., not significant.

Figure 6. Numerosity estimation (study 2). General numerosity estimation performance for each tested numerosity in the (a) **human numerosity estimation task NEH** and (b) **object numerosity estimation NEO** task (study 2). Each dot indicates the individual **human numerosity estimation task NEH** mean (~~over the trials~~) estimate at the corresponding tested numerosity. The dots with the bar on the right sides indicate the in-between subject mean at each presented numerosity. Note the general overestimation bias in **human numerosity estimation task NEH** and **object numerosity estimation task NEO**. Error bar represents 95% confidence interval.

b

b

Figure 7. NEH Human and object numerosity estimation tasks and NEO for both PD patient groups (PD-PH and PD-nH) (study 2). (a) Performance is shown in PD patients for each tested numerosity in the human numerosity estimation NEH task for PD-PH (dark blue) and PD-nH (light blue) separately. Each dot indicates the individual human

numerosity estimation task NEH mean estimate for the tested numerosity (PD-PH (dark blue) and PD-nH (light blue)). The dots with the bar on the left and right sides indicate the mixed-effects linear regression between PD-PH (dark blue) and PD-nH (light blue) at each presented numerosity. (b) **Human numerosity estimation task NEH** in PD patients (PD-PH vs PD-nH). Each dot indicates the individual **human numerosity estimation task NEH** mean estimate (PD-PH (dark blue) and PD-nH (light blue)). The dots with the bar on the left and right sides indicate the mixed-effects linear regression between PD-PH (dark blue) and PD-nH (light blue). (c) **Object numerosity estimation task NEO** in PD patients (PD-PH vs PD-nH). Each dot indicates the individual **object numerosity estimation task NEO** mean estimate (PD-PH (dark red) and PD-nH (light red)). The dots with the bar on the left and right sides indicate the mixed-effects linear regression between PD-PH (dark red) and PD-nH (light red). Error bar represents 95% confidence interval. ** $P \leq 0.01$. **PD = Parkinson's Disease; PD-PH = Parkinson's Disease patients with Presence Hallucination; PD-nH = Parkinson's Disease patients with no Hallucination.**

Concordant changes have been made in supplementary material (Figures S1-S13 ; Table 1 ; supplementary material pages 34-46)

Similarly, the term “digital humans” is confusing, partly because in numerosity studies digits usually mean symbolic numbers, partly because digital (as opposed to analogue) usually means discrete, quantized. I do not know in what sense they are digital, other than they are computer-generated, and computers are digital devices. Call them human images or similar.

We acknowledge the possibility of confusion regarding the use of the term “digital humans”. In our manuscript, the term “digital human” referred to 3D computer-generated versions of human beings, animated to move like real people, in a virtual world. In the original manuscript the term “digital humans” was taken from the terminology commonly used in recent developments of 3D computer-generated humans (Digital humans, MetaHumans, Unreal Engine). However, in agreement with the reviewer's suggestion, we replaced the term “digital humans” with “**virtual human agents**” in the entire manuscript.

More importantly, but along the same lines, I would have appreciated some sort of theoretical explanation of what is going on. I understand the main issue of this manuscript is translational, and there are probably other more theoretical papers in the pipeline, but some theory would aid understanding. Where do the hallucinations come from? – some variant of the predictive coding ideas out there? Is it the case that the hallucinations blend with the human figures,

leading to over-estimation? I presume something like this must be happening, but some discussion, without pre-empting future publications, would be appreciated here.

We have added more details on this to the revised manuscript. Based on data in robot-induced presence hallucination in healthy participants and spontaneous presence hallucination in neurological patients, presence hallucination have been linked with self-related processes, especially sensorimotor signals (Arzy et al., 2006; Blanke et al., 2014) and related sensory prediction signals (i.e., Orepic et al., 2023). Based on the data from the present study (on numerosity estimations) we argue that latter mechanisms interact with those of numerosity estimation and predominantly with numerosity estimation for humans. We have just completed an independent neuroimaging study (using high density EEG; analysis ongoing), in which we investigate whether the brain mechanisms for the numerosity estimation for humans (compared to numerosity estimation for objects) share brain mechanisms of robot-induced presence hallucinations. However, EEG data analysis is ongoing. We have extended the revised discussion (manuscript page 22) on this topic as follow:

“Robot-induced presence hallucinations have been associated with a perturbation of sensorimotor self-related prediction signals (Blanke et al., 2014, Bernasconi et al. 2022, Orepic et al., 2023). That is, our robotic system creates a spatial mismatch between participants’ right arm movements and their sensory consequences (tactile feedback) on the back. Combined with a temporal mismatch (asynchronous sensorimotor condition), this mismatch is resolved by participants perceiving the touch as originating from an external source (another agent, the hallucinated presence). When testing human numerosity estimation, we argue that the experimentally induced ‘presence’ modulates the number of humans estimated to be in the virtual room, as if the ‘presence’ is added to the seen humans. Robot-induced presence hallucinations in healthy participants have been associated with activation of primary motor cortex, somatosensory cortex, premotor cortex and superior parietal lobule, bilateral supplementary motor area, and inferior parietal cortex. Similarly, spontaneous presence hallucinations in neurological patients have been associated with the temporoparietal cortex (Arzy et al. 2006) and frontoparietal cortex (Brugger, Regard, and Landis 1996; Blanke et al. 2014). Interestingly, temporoparietal cortex (Ionta, Gassert, and Blanke 2011), and frontoparietal cortex (Ehrsson, Holmes, and Passingham 2005; Petkova et al. 2011) are key areas of bodily self-consciousness and are areas that integrate sensorimotor or multisensory bodily signals, as shown in human (Serino et al. 2013) and non-human primates (Graziano, Cooke, and Taylor 2000; Lriki, Tanaka, and Iwamura 1996). Neural recordings in monkeys showed that the neural activity of bimodal visuo-proprioceptive neurons located in these regions were

modulated when seeing body parts, but not when seeing objects (Graziano, Cooke, and Taylor 2000; Graziano 1999; for review see Blanke 2012; Blanke, Slater, and Serino 2015). Similar selectivity has been observed in humans (Makin, Holmes, and Zohary 2007; Brozzoli et al. 2011). Numerosity estimation has been associated with similar brain regions (especially with the intraparietal sulcus) (Piazza et al. 2004; Harvey et al. 2013; Nieder 2016). Based on these findings we suggest that the selective overestimation in the human numerosity estimation task, but not for objects (in both healthy controls and PD patients) is due to the recruitment of human body-specific activations overlapping with those of human numerosity estimation. Future imaging work, using numerosity estimation and robotically induced presence hallucinations, should investigate this and validate these hypotheses.”

Finally, please put equality lines on the graphs, to make the over-estimation more obvious (there are symbols, but these get confused with data).

As suggested, we have modified the graphs by adding equality lines (see above response to the first point of reviewer #1 with the modified figures).

All that said, I think the work is great! With very minimal revision, it should make a fine contribution to Nature Communications.

We thank again reviewer #1 for the positive appreciation of our manuscript and constructive remarks, which we believe allowed us to considerably improve the quality of the manuscript.

Reviewer #2 (Remarks to the Author):

The paper by Albert et al. reports a study on 28 healthy controls and 118 PD patients, focused on Presence Hallucinations (PH), and on the possible link between PH and Numerosity Estimations of Human (NEH) figures in a Virtual Environment.

The statistical power of the study is adequate, the applied statistical corrections are state of art, the paper is a further evolution of several papers by the same authors, all published in highly respected journals, where also fMRI evaluations were obtained.

We thank reviewer #2 for the positive appreciation of our manuscript.

Shortly the study shows that :

- 1) In a Virtual Environment (eye goggles), when “Healthy Participants” are operating a virtual robot, and are involved in NE task, NEH errors (with increments as compared to actual stimuli) are reported. The NE do not increase if stimuli are not human figures.
- 2) These errors are furtherly increased if a distractor is employed : a mechanical arm poking on the back of the “healthy participant” after a 0.5 s delay from visual stimuli, poking has no effect if it is simultaneous with visual stimuli, it has no effect also if stimuli are not Human Figures.
- 3) In PD patients the same incremental NEH can be detected in a simpler Virtual Environment, i.e. the patient looking at the task from his home based computer or tablet, without any robotic task or
- 4) These NEH seem to correlate to PH , as reported subjectively, in “Healthy participants” and PD patients.

Extracting this information from the paper is difficult, the presentation is dispersed in repetitive statements, the consistency, in terms of number of participants, is not immediately underlined, it is difficult to understand that Study 1 was only done in controls and Study 2 only in PD patients.

We have made several changes to the revised manuscript in order to further clarify the methods used in both studies, their methodological differences, as well as our obtained results and their presentation. We have also reduced the number of abbreviations.

In study 1 (pre-registered), we used our robotic system to induce presence hallucinations in healthy participants (as reported previously) and report for the first time a selective

overestimation of virtual human agents following the induction of presence hallucinations with a robotic device (asynchronous condition). We motivated the present work on overestimation by referring to prior behavioral-robotic studies from our laboratory, which used a similar experimental protocol to induce presence hallucinations and specific behavioral effects (i.e., Blanke et al. 2014 ; Salomon et al. 2020 ; Bernasconi et al. 2021 ; Orepic et al. 2021; Serino et al. 2021 ; Bernasconi et al. 2022 ; Orepic et al. 2023), as well as a much simpler and uncontrolled human estimation task (Blanke et al., 2014).

In study 2, the logic was different: we extended the numerosity estimation task (as developed in a series of pilot studies and study 1) to PD patients, who suffered from disease-related spontaneous presence hallucinations and investigated the clinical validity of the numerosity estimation measure. For motivating study 2 we also refer to prior but different work from our laboratory, which showed that PD patients who report presence hallucinations as part of their disease in daily life are more sensitive to the robotic induction of the presence hallucinations than patients without this hallucination (Bernasconi et al., 2021; Fig. 1, B; see Figure below). The translation from healthy individuals to patients with PD and the evidence that PD patients with hallucinations are (significantly) more sensitive to our methods of assessment of hallucinations directly motivated the current study 2. For study 2, we expected that PD patients with presence hallucinations in daily life would have a larger numerosity overestimation bias, as compared to PD patients without presence hallucinations in daily life.

[REDACTED]

Figure 1 (Bernasconi et al. 2021). Robot-induced presence hallucination in patients with PD. (B) Robot-induced presence hallucination in patients with PD (asynchronous versus synchronous stimulation).

In addition, as suggested by reviewer #2, we have revised the introduction of the manuscript by adding the sample population and sample sizes for both studies (manuscript pages 5 and 6):

“We combined VR with the robotic system and determined whether the new task is an implicit, quantitative and behavioral marker for robot-induced presence hallucinations, in a group of 28 healthy participants.” ; “... To test this, we ~~next~~ developed a home-based online numerosity task with digital humans-virtual human agents and tested investigated a large group of 170-PD patients with PD at their home without robotic stimulation (study 2).”

The authors make reference to their previous studies in PD patients, in order to support the logic of the study presentation, but my feeling is that supporting with prior studies, in this case, is not methodologically acceptable.

We hope that this concern has been clarified in the previous response. Please also see our next response, which is also related to this point.

While I cannot challenge the appropriateness of methods of data collection and statistical evaluation, I found that the design of the study and its conceptualization has some flaws.

Design :HC

1) Why the set of experiments is not the same for the different groups (Healthy Controls-HC, PD with PH, PD without PH)? We do not know if HC are prone to PH also in a familiar environment, or it is only in the secluded virtual reality of eyegoggles that PH occurs.

This must not have been not clear enough in the original manuscript; we hope that our responses to the concerns above and the related changes we made in the revised manuscript have clarified this point.

In more detail, in study 1, we tested healthy controls to develop and validate the novel protocol and implicit measure (i.e. human numerosity estimation task) to quantify robot-induced presence hallucination, without the possible confounds of verbal reports or confounds associated when presence hallucinations result from disease (e.g. Rogers et al. 2021). Because presence hallucinations are not common in healthy individuals, we have developed the robotic procedure (Blanke et al. 2014 ; Bernasconi et al. 2022) that allows us to induce and study the brain mechanisms of presence hallucination in the laboratory in healthy subjects: in the present study 1 we tested whether robotically-induced presence hallucinations are associated with an overestimation of virtual human agents (in our human numerosity estimation task).

As stated in our response to the previous comments, we also used the numerosity task in study 2, but did not require the use of the robotic system in this study. We expected that the

trait of the PD patients (i.e. having or not having hallucinations) was sufficient to result in a modulation of the numerosity estimation (based on previously published data: i.e., Bernasconi et al., 2021; also see previous response to this point and the figure above); therefore, the use of the robotic system was not required. Moreover, it would have been impossible to use a robot and test 170 individuals, because we would have had to transport and install the robot in the homes of all tested participants. One of the main advantages of the present (and other) online studies is the possibility to test a large number of patients more quickly and avoid that patients have to travel.

We have added the differences in study design (related to VR) to the limitation section of the revised manuscript (manuscript page 25) as follow:

“Eighth, study 1 used 3D VR to present stimuli and study 2 used 2D screens. As 3D VR (study 1) is a more ecologically valid approach (for testing the perception of humans) as compared to a 2D representation (study 2), future work should test PD patients in VR, which may potentially lead to stronger effects.”

In addition, we note that we did not investigate whether healthy controls with spontaneous presence hallucination would have higher human numerosity estimation as compared to healthy controls with no such hallucinations. However, it is an avenue worth exploring and we are currently planning such a study. We have added this to the limitation section of the revised manuscript (manuscript page 24):

“Sixth, while we tested our online human and object numerosity estimation tasks in PD patients, future work should explore whether healthy controls experiencing spontaneous presence hallucinations also exhibit a human overestimation bias as compared to healthy controls (not experiencing such hallucinations).”

2) Why is not poking performed also in the home based experiment? Knowing about the effect of distractors in familiar or unfamiliar environments would add relevant information. .

Please see our previous responses to this point. In addition, we would like to remind the reviewer that the robotic system allows to induce transient hallucinatory states which are similar to what can be observed in patients with this specific hallucinatory trait. The robotic system capitalized on sensorimotor conflicts to induced hallucinatory states in healthy participants (e.g. Blanke et al., 2014), under controlled experimental conditions. The possibility to induce a hallucinatory state in healthy participants allowed us to assess whether numerosity estimation is a valid metric for experiencing such aberrant perception, without the possible confounds of disease. To clinically validate numerosity estimation we tested the same task in

patients with the trait of having or not hallucinations. However, in this clinical population it was not necessary to induce the hallucinatory state because they were already characterized by the trait of having presence hallucinations. That is, at the home-based experiment we aimed at investigating whether the trait of having presence hallucinations would result in a modulation of the numerosity estimation.

We have amended the introduction section of the manuscript as follow (manuscript page 6):

“Based on these results and our previous finding that PD patients with minor symptomatic presence hallucinations (compared to PD patients without those hallucinations) show heightened sensitivity to robotically induced PH presence hallucinations (independent of asynchronous stimulation) (Bernasconi et al. 2021), we hypothesized that PD patients with symptomatic presence hallucinations (PD-PH) would show an overestimation for virtual human agents as compared to patients with PD but without hallucinations (PD-nH). To test this, we next developed a home-based online numerosity task with digital humans virtual human agents and tested investigated a large group of 170 PD patients with PD at their home without robotic stimulation (study 2).”

And the introduction section of study 2 of the manuscript as follow (manuscript page 13):

“Would performance of PD patients with PH presence hallucinations as part of the disease be characterized by an increase in human numerosity estimation NEH, compared to those PD patients without PH presence hallucinations, even without any robotic stimulation? Recent work adapting the robotic presence hallucination induction paradigm to PD indicated that PD patients experiencing symptomatic presence hallucinations PH (PD-PH) had a six-fold higher sensitivity to the robotic sensorimotor procedure as compared to PD patients who never had PH presence hallucination (PD-nPH), suggesting that experiencing symptomatic presence hallucinations results in a bias in experiencing robot-induced presence hallucinations (Bernasconi et al. 2021). In combination with the results from study 1, these clinical data suggest that (1) PD-PH patients may have a bias in human numerosity estimation task NEH, that (2) this bias should exist without being exposed to robotic stimulation, and that (3) such a human numerosity estimation task NEH bias should be larger than the one in PD patients without such hallucinations (PD-nH).”

Distractors/familiarity: Both mechanisms cannot explain the present results on numerosity estimation and presence hallucination. Thus, previous research and literature shows that experiencing presence hallucinations results from a misattribution of one’s own bodily signals (e.g. Arzy et al., 2006; Blanke et al., 2014, Bernasconi et al., 2021). The robotic device used in study 1 is not used as a distractor, but allows the induction of presence hallucinations in the asynchronous sensorimotor condition (and not in the control synchronous sensorimotor

condition). Nor is virtual reality used as a distractor. In addition, the numerosity estimation task is performed after (not during) the robotic sensorimotor stimulation. Finally, the numerosity estimation effect has been shown to be selective for humans and not for control objects in study 1 and 2. Based on these data, it is very unlikely that mechanisms of distraction/familiarity explain the present results.

Conceptualization:

Presence Hallucinations (PH) are a small aspect of hallucinations in PD; with passage hallucinations and illusions, PH are considered the typical triad of “early” hallucinations, i.e. occurring early in the course of PD, as can be found in papers quoted in the present study. Several descriptions show that early hallucinations can wane, to reappear later in advanced PD, shortly followed by complex hallucinations. One of the studies quoted in the present paper (Bejr-Kasem et al.) shows that a Default Mode Network disruption can be observed in early hallucinators, and it is similar to what observed in late -hallucinators, but the clinical aspect and the insight are extremely different, the amount of distress makes the difference, and the relevant therapeutic target is likely to be the distress linked to complex hallucinations, as it is well known that the early hallucination wanes when the PD patient focuses his attention to it. Therefore the first question would be: given the clinical relevance, why was PH considered an important target?

We thank reviewer #2 for this question. In previous studies, presence hallucinations have been reported as one of the most common minor hallucinations in PD (together with passage hallucinations) and they are usually experienced at early-mid stages of PD (Ffytche and Aarsland 2017; Lenka et al. 2019). We also note that minor hallucinations can in some patients even precede parkinsonian motor symptoms (Pagonabarraga et al. 2016). Importantly for our study, recent data also show that minor hallucinations (presence and passage hallucinations) are not only among the earliest hallucinations occurring in PD, but that they also share brain alterations with structured visual hallucinations (Pagonabarraga et al. 2014; Helena Bejr-kasem et al. 2019) and are linked to more rapidly developing cognitive deficits (Bernasconi et al. 2023; H. Bejr-kasem et al. 2021; Bernasconi et al. 2021). This underlines their potential role as an early marker for dementia (Pagonabarraga et al. 2014; H. Bejr-kasem et al. 2021; Bernasconi et al. 2021; 2023). Accordingly, we consider them an important symptom to investigate and better understand hallucinations in PD. This has been further clarified in the revised manuscript. We have amended the discussion section of the manuscript as follows (manuscript page 20):

“In study 2, we successfully translated NEH the human numerosity estimation task measurements to an online assessment of hallucinations performed by elderly PD patients, who carried out the task on their personal computer or tablet at home. Presence hallucinations are an important symptom in Parkinson's disease, as they are usually experienced at early stages of the disease (Ffytche and Aarsland 2017; Lenka et al. 2019), may even precede Parkinsonian motor symptoms (Pagonabarraga et al. 2016), and are linked to more rapidly advancing cognitive decline (Bernasconi et al., 2023). The present data Data show that PD patients with disease-related presence hallucination PH in their daily life have ...”

Methods to induce increments of hallucinations were already described, i.e. Pareidolia tests for Illusions (Uchiyama et al. Brain 2012) and Virtual Environment for Early and Late Hallucinators (Onofrj et al. J Neurol Sci 2006) We know, from these studies, that a virtual environment increases hallucinatory percepts in PD, in early and late hallucinators.

Thus the second question is :why was not the effect of Virtual Reality compared in other tasks?

This is a very interesting point, and we are aware of the cited studies on complex visual illusions and visual hallucinations. However, in the present study 1 we are not employing VR to induce hallucinations, nor investigate whether VR versus real life conditions lead to differences in the frequency of hallucinations. Instead, we used VR in order to expose participants in a more ecologically valid way to our visual numerosity stimuli (see our response to point 1 of design in healthy participants) and to measure overestimation biases. To induce presence hallucinations, we are using our existing and well-described procedure that uses the robotic device (Blanke et al. 2014 ; Salomon et al. 2020 ; Bernasconi et al. 2021 ; Orepic et al. 2021; Serino et al. 2021 ; Bernasconi et al. 2022 ; Orepic et al. 2023). We also note that robotic induction was always carried out prior to the presentation of our numerosity stimuli in VR. Study 2 did not use VR and only used standard non-VR equipment that patients had at their homes. It may be interesting in future studies to investigate whether PD patients not only report more visual hallucinations when immersed in a VR environment (Onofrj et al. 2006, Albani et al. 2009, Albani et al. 2015), but also more presence hallucinations. However, this was not tested in the present study 2.

Furthermore, although some measures exist to assess visual illusions and hallucination (as mentioned by reviewer #2, for example pareidolias), there are no existing quantitative measures for presence hallucination (which is yet only assessed through questionnaire or clinical interviews). In the present manuscript, we are proposing and validating the first implicit measure of presence hallucinations (i.e. our human numerosity estimation task).

Finally, in their introductory claim, the authors state that assessment of hallucinations is biased by verbal report, as obvious, but the correlation of NEH with PH is again only based on a verbal, subjective, report by the patient, or control, who says he/she had PH.

To me, this is the major logical challenge. More solid correlation of NEH and hallucinations could be obtained, in example, by selecting patients on the basis of fMRI alterations.

We agree that neural markers as suggested by reviewer 2, are even less direct than verbal or behavioral markers (as neural markers do not require any response from the patient). Our distinction, leading to the present investigation is that there is a difference between markers based on a verbal response by the patient (associated with many biases) and an implicit behavioral response (i.e., reaction times, accuracy, overestimation bias) that requires a button press or number estimate. Our present data show that human numerosity estimation provides an implicit and more quantitative measure of presence hallucinations, without any verbal rating of the hallucination (in study 1, participants move their head to target an answer and validate by saying 'ok'; in study 2, PD patients enter their answer in a free-text input field). Our ongoing and unpublished research complements these findings on numerosity estimation and aims to define neural markers (using high density electroencephalography and the same task; analysis ongoing).

We have extended the revised discussion (manuscript page 23) on this topic as follows:

“Future imaging work, using numerosity estimation and robotically induced presence hallucinations, should investigate this and validate these hypotheses.”

Minor:

line 432- “elderly PD patients” emerge from nowhere in the study. When was an age stratification performed?

We corrected the formulation to **“PD patients”**.

Line 174 and 348 –“subitizing “ is an unusual English word, mostly used in psychology studies, the authors should explain it, and should also say why they believe that subitizing for numbers above 5 is acceptable.

We thank reviewer #2 for this suggestion, and have added the definition of subitizing to the manuscript (manuscript page 8):

“This behavior is typical and has been observed with dots stimuli for numerosities just above the subitizing range (Fornaciai and Park 2020; Jevons 1871; Minturn and Reese 1951). The

subitizing range corresponds to fast, accurate and confident number judgements, only observed for a low number of dots or items. Above this range, the number of items can be either counted accurately but more slowly or estimated rapidly but with errors.

We believe that subitizing for numbers above 5 is acceptable following our online pilot study described in the manuscript as follow (manuscript page 8):

“To determine the lower bound of the estimation range of our **human numerosity estimation task NEH** stimuli and select the range of presented numerosities in study 1, we conducted an online pilot study ~~(see Note S1)~~ in 28 healthy participants **(see Note S1)**. In this online preliminary study, participants were asked to indicate the number of **virtual human agents** they perceived in flashed human stimuli over a broad range of numerosities (ranging from 1 to 24). This online preliminary study indicated the lower bound of the estimation range of our **NEH stimuli human numerosity estimation task** stimuli to be 5. For all methodological aspects and detailed results of this online pilot study see Note S1.”

In this online pilot study (detailed in supplementary note 1), we found the lower bound of the estimation range of our human numerosity estimation task to be 5 (and so the subitizing range to be 1-4 virtual human agents, similar to what is usually described in literature on more simpler stimuli as dots or squares). As no errors are expected in the subitizing range, we selected our lower range to be 5 (just above the subitizing range). We have amended the introduction of supplementary note 1 to highlight this message (supplementary material page 2):

“An online web-based experiment was developed to determine the subitizing range of **human numerosity stimuli, and thus select the range of stimuli to display in study 1. We aimed at selecting a range of stimuli in the early range of numerosity estimation process (just above the subitizing range), where numerosity estimation errors start to occur.**
Thus, when the presence hallucination is induced, this felt presence (which is not visual) would interact with the uncertainty of numerosity estimation of virtual human agents and push the estimation system of human agents toward the upper number. This would not be the case when boxes are shown.”

Suggested readings :

Collerton D et al. Neurosci Biobehav Rev, 2023

O'Brien D et al. J Neurol Neurosurg Psychiatry, 2020

Onofrj et al. Mov Dis, 2015 and 2019

We thank again reviewer #2 for the positive appreciation of our manuscript and for the various comments. We hope we have been able to clarify our protocol and results.

Reviewer #3 (Remarks to the Author):

Review for “Digital-robotic markers for hallucinations in Parkinson’s disease”
in Nature Communications

Summary: in this article, the authors present two studies. The first demonstrates in young, healthy patients the usefulness of a robotic protocol combined with virtual reality (VR) to i) induce presence hallucinations (PH) and **ii) study the association of the latter with numerosity estimation (NE) of virtual human agents**. The second presents a study carried out remotely by Parkinson's patients, with or without PH, and attempts to show the association between the latter and the NE of virtual human beings. Together, these two studies attempt to demonstrate the efficacy of NE as an objective and quantitative marker of PH, while presenting a remote protocol supported by laboratory results.

We thank reviewer #3 for the positive appreciation of our manuscript and the methods used, along with the different valuable suggestions, which we believe have helped us improve the manuscript.

Disclaimer: as I'm not an expert in Parkinson's disease, I can't assess the theoretical relevance of evaluating and quantifying PH. I will focus on my field, which corresponds to the virtual experimental protocol, notably on the methodological aspects of VR and the extrapolation of results, as well as on the sense of (co)presence.

General criticism: both the experimental and analytical methodology used in these two studies appear to be rigorous. The demonstrative reasoning is clear and follows logically from the results presented by the two studies, although I find that the presentation format would benefit from being a bit more orthodox.

We acknowledge the suggestion about the presentation format, and have revised the manuscript accordingly with several modifications that, we believe, will facilitate the reading of the paper. For example, the number of abbreviations used has been highly reduced; abbreviations have been completely removed from figure legends, titles, and captions. We changed the term “digital humans”, which could lead to confusion depending on the research field, to “**virtual human agents**”. We added some definitions in the manuscript as for the

subitizing range. All modifications are indicated as track changes in the revised version of our manuscript.

If fellow reviewers who are experts in the theoretical and clinical field endorse the paper's publication by raising its relevance, I certainly won't object. However, I think the paper could gain in rigor by clarifying the points I am now describing. Indeed, I find that there are points of importance that are not discussed in the article and should be:

- 1. The NE stimulus is sometimes indicated as lasting 200ms and sometimes 250ms. I don't think this difference is justified in the manuscript. Please clarify whether this is a drafting error or whether there really is a difference, and if so to justify it.

We thank the reviewer for raising this point. There is indeed a difference in stimulus presentation time between study 1 (200ms ; young healthy participants in VR) and study 2 (250ms ; elderly with PD on their computer or tablet). These values have been selected based on previous literature and own pilot data. That is, in the numerosity literature, stimuli for numerosity estimation tasks are mostly presented for a reported duration of 200ms or 250ms (and in some rare studies up to 300ms). When designing study 1 (that targeted young healthy individuals, we choose a stimulus presentation time of 200ms as most commonly used in numerosity studies. This also corresponded as an appropriate number for display in the HMD, because Oculus Rift CV1 ran at 90Hz; 18 frames were rendered in precisely 200ms). In study 2, our target population changed to PD patients, who are also much older and can be cognitively impaired, compared to our healthy sample of study 1. During pilot testing (development of the task for study 2, carried out in healthy elderly participants), we observed some difficulties in performing the task with a stimulus presentation time of 200ms, but not (or less) at 250ms. We thus decided to use 250ms for study 2 (also because this duration is also widely used in numerosity studies and, additionally, corresponded to 15 or 30 frames on widely used 60 and 120Hz computer or tablet monitors that we expected patients to use at their homes (study 2)).

The method section of study 2 has been amended as follow (manuscript page 36):

“The stimuli were displayed for 250ms (Cicchini, Anobile, and Burr 2014; Fornaciai and Park 2020; Birren and Botwinick 1955). **The increase in stimulus presentation duration from study 1 (200ms) to study 2 (250ms) was based on pilot data in elderly healthy participants, adapting the difficulty of our human and object numerosity estimation tasks from young healthy participants (Study 1) to patients with Parkinson's disease (Study 2).**”

- 1. I fully understand the interest in using VR in this context, and I think it could quite possibly make the task more discriminating and relevant, by appealing to ecological cognitive mechanisms for detecting people (e.g. 3D movements, posture...) that are much harder to trigger in 2D. However, reading the article and the reasoning behind it suggests a logical continuation, a transposition of Study 1, in VR, to Study 2, on computer/tablet. It seems to me that these two media involve different cognitive and motor perceptual processes. However, these differences are not discussed in the article when articulating Study 1 and Study 2. I believe that a discussion on this subject could greatly enrich the manuscript. For example, in Study 1, NE was reported using a gauge to assess the ordinal distance between numbers, whereas in Study 2 it was an open-ended question with no visual indication. I don't have any empirical references of an effect on this specific subject, but I think studies somehow intended for comparison should try to follow a similar methodology on these kinds of aspects. I think this (and other aspects of comparison) deserve a point of discussion.

We thank the reviewer for the comment and agree with the suggestion. We indeed believe that the use of 3D VR, creating a more ecologically valid environment than a 2D screen, could lead to higher effects compared to the ones observed in study 2 in PD patients. We have amended the limitation section of the revised manuscript as follow (manuscript page 25):

“Eighth, study 1 used 3D VR to present stimuli and study 2 used 2D screen. As 3D VR (study 1) is a more ecologically valid approach (for testing the perception of humans) as compared to a 2D representation (study 2), future work should test PD patients in VR, which may potentially lead to stronger effects.”

In addition, the response mechanism of study 1 (slider and voice recognition, healthy controls) was adapted for the new targeted population (PD patients) and set-up (online setting, 2D screen). We designed the answer mechanism of study 1 (VR) with the goal to reduce a potential break of presence that could have been triggered by interacting with an external physical device, and thus came with this slider and voice recognition strategy (we technically tested different answering mechanism, especially full number voice recognition which was not enough robust across individuals). This was also an extension of the answer mechanism introduced and used in study 1 to provide answers for the questionnaire (slider and voice recognition). The maximum number on the displayed scale was chosen based on pilot data described in Note 1 and pilot test data of our experiment, for numerosity stimulus between 5 and 8. In study 2 (online setting, 2D screen, PD patients), we opted for standard ergonomics interaction mode for online report of a number (textbox accepting only numbers).

- 3. You compare the differences in reaction times (not significant) to say that “Additional analysis ensured that the observed differences between NEH and NEO are not explained by differences in task difficulty.”. I think it's a bit bold to assert such a thing on the sole basis of reaction times between conditions, all the more so for a response based on a very short stimulus (200ms) that no longer exists at the moment of response. What's more, I'm not convinced that reaction time is sufficient to say that there is no difference in difficulty between the two tasks. I invite the authors to nuance their remarks here and add a quick discussion on the subject. For example, it seems to me that the bright green of the control condition could make the task easier, compared with the dark colors of the individuals. This could well moderate or modulate performance.

We have modified our discussion of this point. The corresponding sections of the manuscript have been revised as follow (manuscript page 11):

“Additional analysis ~~ensured that the observed differences between NEH and NEO are not explained by differences in task difficulty. Thus, we observed~~ revealed no significant differences in response times between ~~NEH and NEO human and object numerosity estimation tasks~~ (i.e., type of stimuli; $F(1,2197)=0.73$; $p=0.39$; no main effect plus no interactions; Table S4) (Figure S3; Figure S4; Figure S6; Figure S7). This analysis also indicated no effect of robotic sensorimotor stimulation on response times (i.e., type of robotic sensorimotor stimulation; $F(1,2197)=3.01$; $p=0.08$; no main effect, no interaction; Table S4), suggesting that the different robotic sensorimotor stimulation conditions did not affect numerosity estimation task difficulty nor alertness. This is supported by the fact that the numerosity estimation task is not performed during, but just after the robotic stimulation. ~~These results consolidate our findings that the overestimation observed in the asynchronous versus synchronous condition and in NEH versus NEO cannot be explained by differences in task difficulty.~~”

And (manuscript page 12):

“Critically, this effect was observed (1) when comparing the ~~PH-presence hallucination~~ inducing asynchronous ~~sensorimotor~~ condition with the synchronous ~~sensorimotor~~ control condition, ~~and~~ (2) was absent in the ~~object numerosity estimation NEO task, and (3) cannot be explained by differences in task difficulty or alertness.~~ Observing that ~~(4)(3)~~ the magnitude of the ~~human numerosity estimation task NEH~~ bias, but not the ~~object numerosity estimation task NEO~~ bias, correlates with ~~PH presence hallucination~~ ratings, further links the ~~human numerosity NEH~~ overestimation with ~~PH presence hallucination.~~”

And (manuscript page 17):

“Additional analysis revealed no statistical differences in response times between the two PD groups (i.e., PD group; $F(1,116) = 0.58$; $p=0.45$; no main effect nor any interaction; Table S11) (Figure S8; Figure S9; Figure S11; Figure S12), showing suggesting that task difficulty (online ~~NEH and online NEO~~human and objects numerosity estimation tasks) did not differ between both PD groups (PD-nH and PD-PH). ~~These results consolidate our previous findings and show that the overestimation observed in PD-PH versus PD-nH and in the online NEH versus online NEO cannot be explained by differences in task difficulty between groups.~~”

And (manuscript page 19):

“Importantly, this effect is absent for object numerosity estimation task NEO (Figure 3c) and cannot be explained by differences in task difficulty or attentional processes.”

And (manuscript page 20):

“This effect was absent in the online object numerosity estimation NEO task and cannot be explained by differences in task difficulty between groups, nor does not it depend on clinical covariates such as age, gender, disease duration, or first affected side, corroborating data from study 1, extending them to a large clinical cohort of PD patients, and underlining human numerosity NEH overestimation as a digital marker of PH presence hallucinations in PD.”

And (manuscript page 20):

“This effect is absent for the online object numerosity estimation task NEO assessment (Figure 7c), cannot be explained by between-group differences in task difficulty, and does not depend on any of the acquired clinical covariates (i.e., age, gender, disease duration, first affected side).”

Moreover, we piloted the object numerosity estimation task in VR to match response times and accuracy of the human numerosity estimation task in VR. During the object numerosity estimation task, the edges of the boxes were also brownish (as shown in supplementary figure 1, supplementary material page 41), replacing the green edges present in normal lightning condition during the habituation phases (highly reducing the salience of the edges and closer to the colorimetry of humans). In study 2, we generated stimuli from the 3D VR human and object numerosity estimation tasks, in 2D, with enhanced lightning (see supplementary figure 1, supplementary material page 41), justified by the new targeted population (PD patients, an older population than study 1 population, and known to have visual disturbances such as reduced contrast sensitivity).

- 4. If I'm not mistaken, you give very little information on head movements in virtual reality. What were the instructions during habituation phases with humans or objects? Were participants free to move? Did they have to look straight ahead all the time? Was it natural to look straight ahead, as there was nothing to look at on the sides or behind? This needs further clarification. Head rotations can have significant effects in virtual reality, particularly on cybersickness. I think it would be a good idea to add a word about cybersickness to the method. Did any participants complain?

Although we did not tell our participants explicitly that they could look around (neither told them not to) during the habituation phase, the design of our experiment made it natural for them to do so during the instruction phases and habituation phases; most participants did so spontaneously upon their first immersion, looking around the virtual room that was designed to match the real one. Then, the virtual experimenter was located on the left side during the instructions, implicitly inviting participants to move their head toward him when giving the instructions. The robot training was more centered, inviting participants to look at it when it appeared. Finally, the virtual experimenter leaves the virtual room through the virtual door, and virtual human agents and boxes are seen as arriving and leaving the virtual room through the door. Participants naturally follow them with their head and get attracted to where things happen (virtual human agents/objects moving, hearing virtual human agents' footsteps).

We have amended the method section of study 1 as follow (manuscript page 31) to clarify head movements during habituation phases:

“Each block started with a 60sec habituation phase, **during which** several **virtual human agents** moved and discussed in the virtual environment in the digital human part (around 10 in total per habituation phase) **(Video S3, Video S4). Virtual human agents could navigate in the whole virtual room, sometimes passing on the side or behind the participant. Participants were implicitly invited to look around and observe the whole virtual environment during these habituation phases.**”

We acknowledge the difficulty of translating to a 2D video what participants experienced in virtual reality and have noticed that presenting videos from a fixed point-of-view at the position of the participant for the habituation phases was a mistake as that doesn't accurately reflect the participant's experience. We have thus generated a novel video of the direct view inside VR of a participant during a habituation phase, where we can especially observe the participant's head movement and synchronized VR view (see video S3).

Concerning cybersickness, participants were only instructed prior to the experiment not to move from the chair where they were seated. As participants were seated during the experiment, and not spatially displaced within the real room nor in the virtual environment, and because head rotation and position were tracked and translated into the virtual environment accordingly, we did not expect our paradigm to induce strong cybersickness. Cybersickness, and more precisely symptoms and effects induced by virtual reality, were assessed in our experiment by a questionnaire administered at the very end of the experiment (for Study 1) (i.e. this is done systematically for all VR studies in our Lab). Our participants did not report cybersickness. This was not detailed in the main text of the manuscript and has been referenced in the method section and detailed as a supplementary note (supplementary note 9). The manuscript method section of study 1 was amended as follow (manuscript page 27):

“At the end of the experiment, participants VR related experiences were assessed with a questionnaire about typical symptoms and effects induced by the experiment and virtual reality (note S9).”

And the supplementary note 9 was added to supplementary material (supplementary material page 21):

“Supplementary note 9: Symptoms and effects induced (study 1)

At the end of the experiment, once the VR headset was removed, participants were assessed with a questionnaire about symptoms and effects induced by the experiment and virtual reality. This questionnaire contained 5 items (“Did you feel nauseous?” ; “Did you feel headaches?” ; “Did you feel dizzy?” ; “Did you feel tired?” ; “Have you experienced any loss of balance?”) rated on a 7-item Likert scale. 1 corresponds to “Extremely intense feeling”, 2 to “Very intense feeling”, 3 to “Intense feeling”, 4 to “Moderate feeling”, 5 to “Weak feeling”, 6 to “Very weak feeling”, and 7 to “Absent”. In addition to these 5 items, participants were presented with one text zone of free report on “Other symptoms / Notes / Suggestions”.

Question	Mean	SD	95% CI lowe r	95% CI uppe r	Corresponding feeling
Did you feel nauseous?	6.79	0.5	6.59	6.98	Absent

Did you feel headaches?	6.25	1.11	5.82	6.68	Very weak feeling
Did you feel dizzy?	5.93	1.15	5.48	6.38	Very weak feeling
Did you feel tired?	3.93	1.09	3.51	4.35	Moderate feeling
Have you experienced any loss of balance?	6.57	0.63	6.33	6.82	Absent

In addition, 6 of the 28 total participants tested made a remark in the “Other symptoms / Notes / Suggestions”:

- “For headaches, lower now that I have removed the headset”
- “Pain in my neck”
- “Back ache, due to the sitting down position”
- “Arm ache, back ache (due to the moving of the arm)”
- “I had pain in the neck during the experiment due to the weight of headphones”
- “Neck stiffness”
- “Nice massage”

Thus, few participants reported tiredness (moderate feeling – likely due to the long duration of the experiment), neck stiffness and back ache (generated by the sitting position and weight of the head mounted display over time).”

We also added the following figure to the supplementary note (supplementary material page 22):

“

Figure Note 9. Participants responses to symptoms and effects induced questionnaire reported in study 1. Each dot indicates a participant’s answer. The dots with the bars on the right side indicate the in-between subject mean for each question. Error bars represent 95% confidence interval.”

- 5. When watching habituation videos with humans, some virtual agents come and stand behind the participant's angle of view, so that they disappear. With videos, it is very difficult to determine whether this effect also exists in virtual reality, which could lead to head movements (see previous point). However, depending on the instructions, if virtual human beings pass behind the subject and the latter has no opportunity to turn around, it is quite possible that this will have an impact on the perception of Presence Hallucination and co-presence (especially considering the tactile stimuli behind the participant's back). Please clarify this point

We hope we have been able to clarify this point with our response to point 4. Indeed, during the habituation phases, some virtual human agents come and stand behind the participant's field of view. Participants could freely move their head to look around, to observe their surroundings and follow the virtual human agents. While this can possibly have an impact on presence hallucination, this is well controlled in our experimental design with our randomization across participants, and across robotic manipulation condition order (asynchronous (presence hallucination-inducing condition); synchronous condition). It can

hence not explain, for example, differences between asynchronous and synchronous conditions.

- 6. In addition to all this, I'm surprised by the choice of objects in the control condition and their movement animation. Why not choose something closer to human beings? Indeed, the objects: i) appear and disappear rather than move, which greatly complicates their overall apprehension, ii) don't seem to come, one by one, through the front door as humans do in some videos, iii) never seem to pass behind the participant as humans do, iv) never seem to stand still as humans sometimes do. All these points deserve important clarification.

The role of the control condition was to compare overestimation biases when presenting biological living entities (such as a presence hallucination) as compared to objects which are not close in nature to a human 'presence'. We piloted with different control objects (such as 3D extended spheres) and different animations (such as boxes moving), but these were associated with anthropomorphic characteristics and behaviors, not fulfilling the desired role of the control condition. We then piloted the object numerosity estimation task in VR to match response time and accuracy of the human numerosity estimation task in VR, for each presented numerosity. We also note that this was the first study of this kind and are aware that several other control objects could be tested, ranging from scrambled virtual humans to inverted virtual humans to living non-human animals and yet other control objects. These studies will be important for the cognitive neuroscience of visual human perception and how this is potentially modulated by an invisible hallucinated person. However, this would go beyond the present study, which also aimed at translating our findings from healthy controls (study 1) to patients with PD (study 2).

We have amended the discussion section of the manuscript as follow (manuscript page 25):

“Seventh, although we specifically designed the control condition (object numerosity estimation task) with objects that are not close in nature to a hallucinated presence, future studies should design additional control conditions and test numerosity estimation for several other control objects, ranging from scrambled virtual humans to inverted virtual humans to living non-human animals and yet other control objects. These studies will be important for the cognitive neuroscience of visual human perception and how this is potentially modulated by an experimentally induced invisible hallucinated person.”

Moreover, regarding the behavior of boxes as compared to virtual human agents, boxes materialize and dematerialize over time on the path (position and orientation matched) of the virtual human agents. Thus, control objects come from the front door as did the virtual human

agents, sometimes materialize behind the participant or stand still, again as the virtual human agents. Presenting 2D videos with fixed point-of-view for the habituation phases didn't accurately reflect the participant's experience. We have replaced these videos (previously supplementary videos 6-13) with two supplementary videos (supplementary videos S3 and S4) showing the direct view inside VR of a participant during a habituation phase (supplementary videos S3) and a closer comparison between habituation phases with virtual human agents and objects (supplementary videos S4). Especially, the supplementary video S4 shows the different habituation phases, compared side by side between conditions (virtual human agents and objects), from a fixed view in position of the participant (properly captioned) and from a top view perspective, (Video S4).

In addition, we have amended the method section of study 1 as follows (manuscript page 31), in order to clarify the 'behavior' of the virtual boxes in the control condition during the habituation phases:

“Each block started with a 60sec habituation phase, where several virtual human agents moved and discussed in the virtual environment in the virtual human agent part (around 10 in total per habituation phase) (Video S3 ; Video S4). Virtual human agents could navigate in the whole virtual room, sometimes passing on the side or behind the participant. Participants were implicitly invited to look around and observe the whole virtual environment. Some of them (between 2 and 3 depending on the habituation phase) were discussing together. There was a total of four different habituation phases, presented in the same order across participants. **In the control (object) condition, objects (boxes with a wireframe shader) materialize and dematerialize over time on the path (position and orientation matched with those of the virtual human agents) in the corresponding habituation phase (Video S4 shows a direct comparison of virtual human agents and objects habituation phases). During the virtual human agent's habituation phases, the virtual human agents perform a succession of pre-recorded and pre-determined motion captured animations (animations are looped and posed-matched). During the virtual human agent's habituation phases, audio footsteps are spatially rendered at the location and timing of each virtual human agent's foot new contact with the ground. During the corresponding control object habituation phases, a new control object materializes each 3 animations of each virtual human agent at its position and with its orientation. This control object then dematerializes after 5 animations of the corresponding virtual human agent.”**

- 7. I think the general behavior of individuals/objects during the habituation phases deserves more detail. How were their movements decided?

This has been added (manuscript page 31) as described in the response to the previous point.

- 8. Some Post-hoc comparisons are carried out with p-value corrected for multiple comparisons, while others are not. I think this deserves a sentence of justification. In addition, I'm not sure that confidence intervals and effect sizes are systematically reported. Please take the time to check that each statistical test includes, where applicable, the confidence interval and effect size rather than just the p-value.

We thank the reviewer for this comment and have noticed an error in our text. Post-hoc comparisons are not corrected for multiple comparisons. The method section of study 1 was amended as follow (manuscript page 32):

"The significance of fixed effects was estimated with likelihood Ratio Test. Post-hoc analysis was performed on the significant interactions and corresponded in pairwise comparisons using independent-samples t-tests, ~~corrected with Holm's sequential Bonferroni procedure not corrected for multiple comparisons~~. The general estimation performance at each numerosity was assessed with one-sample t-tests against presented numerosity, with reported p-values not corrected for multiple comparisons. The difference of estimation between ~~human and objects numerosity estimation tasks~~ **NEH and NEO** was assessed with ~~two-sample paired sample~~ t-tests at each of the presented numerosity, with reported p-values not corrected for multiple comparisons."

Accordingly, the method section of study 2 was amended as follow for clarification (manuscript page 37):

"The significance of fixed effects was estimated with likelihood Ratio Test. Post-hoc analysis was performed on the significant interactions and corresponded in pairwise comparisons using independent-samples t-tests, ~~corrected with Holm's sequential Bonferroni procedure not corrected for multiple comparisons~~. The general estimation performance at each numerosity was assessed with one-sample t-tests against presented numerosity, with reported p-values not corrected for multiple comparisons. The difference of estimation between human and object numerosity estimation tasks ~~NEH and NEO~~ was assessed with paired sample t-tests at each of the presented numerosity, with reported p-values not corrected for multiple comparisons."

The suggestion of reporting confidence intervals and effect sizes is good. We have amended the manuscript and added confidence interval and effect size to each applicable statistic where it was not initially reported (manuscript pages 10, 11 and 17 ; supplementary material pages 21 and 27).

- 9. It seems to me that the authors do not report any details on the conditions under which the PD participants took part in Study 2. However, this type of study, particularly as it includes a strong dimension of perceived human presence, deserves a word on the conditions under which it was carried out. Did you ask the participants to be alone? Did you control their presence? It seems to me that the number of humans actually present in the experimental room can have a significant impact on the results, and that this merits a point of discussion.

We indeed think the presence of another person in the room can have an impact on the task and on the participants' answer. In this regard, we did indeed instructed participants to perform the task individually in the text accompanying the link to the study (newsletter, website, flyers) as follow: "This study: - is conducted individually (not in a group)".

The method section of study 2 has been amended with the following text (manuscript page 34): "**Participants were instructed to perform the experiment while being alone in the room.**"

However, as the study was carried out online and anonymously, we do not know whether participants were actually alone during the experiment or not (again the object control condition and the reported findings controls for such a general bias). We added the following point in the discussion section (manuscript page 24):

"Fifth, although we instructed PD patients in study 2 to perform the task while they were alone in the room, due to the online nature of study 2, we cannot ascertain that participants were actually alone during the experiment. The potential presence of other persons may have an impact on human numerosity estimation task and requires further investigation."

- 10. You note that "Participants were instructed to stay one meter away from their screen on a computer and fifty centimeters away from their screen on a tablet". You should justify this.

Participants were indeed instructed to stay one meter away from their screen on a computer and fifty centimeters away from their screen on a tablet. This difference in distance, in combination with the same ratio difference for displayed stimuli, permits to have a similar stimulus viewing angle on computer and tablet while maximizing their size (as computers generally have bigger display size than tablets). The exact values (50cm and 1m) were then taken from the US Occupational Safety and Health Administration (OSHA) viewing distance recommendation (20 – 40 inches (50 – 100 cm) - <https://www.osha.gov/etools/computer-workstations/components/monitors>).

We amended the method section of the manuscript as followed (manuscript page 34):

“Screen calibration

Participants were invited to measure and report the length of a line displayed on their screen. This measure ~~allowed to scale stimuli, so they were displayed the same physical size on each participant screen, independent of monitor screen and positions, allows scaling the stimuli so that their physical size on the display is the same for all participants, independently from the physical size of the monitor screen or tablet used by the participant. In addition, in order to approximate a controlled viewing angle for all participants, we adopted a stimuli presentation ratio of 2:1 based on the device used (stimuli displayed on computers were two times bigger than those displayed on tablet), in combination with a viewing distance ratio of 2:1. That is,~~ participants were instructed to stay one meter away from their screen on a computer and fifty centimeters away from their screen on a tablet. ~~These values were chosen based on the US Occupational Safety and Health Administration (OSHA) ergonomics research viewing distance recommendations (from the eye to the front surface of the computer screen, between 20 and 40 inches, i.e. 50 and 100 cm (<https://www.osha.gov/etools/computer-workstations/components/monitors>)).~~ The detection of whether the participants were using a computer, or a tablet was automatic.”

Although we have suggested these distances to participants, we did not control the participant's actual distance from the screen during the experiment. One method we thought of that could have been used is the VirtualChinrest (Li, Joo, Yeatman, and Reinecke, 2020), which allows to estimate viewing distance by detecting the participant's blind spot location. However, this task can be difficult to perform, especially with our sample population (PD patients), would not work on a tablet (eccentricity of blind spot exceeding screen size, tablet being slightly moved and rotated when manipulated), so we chose not to implement this method.

- 11. It seems that authors use abbreviations (e.g. NE) before the first occurrence of the term. Please take the time to check that each abbreviation is explained.

Following other comments about the excessive use of abbreviations in our manuscript, we have carefully revised the manuscript to avoid this problem and have also reduced the number of abbreviations to facilitate the reading of the paper.

- 12. Finally, and although biased by my research, I would amplify in the article's discussion the potential importance of the sense of presence in this task. I'll measure the latter using questionnaires not only of co-presence but also of spatial presence, and assess its impact on participants' performance on the NE task as well as on their feeling of PH.

We agree on the potential importance of the sense of presence (as defined in the virtual reality field) in this task, and thus raised this issue in the initial discussion limitations of the manuscript as follow (manuscript page 24):

“Second, in study 1, participants are immersed in a virtual environment where a virtual character speaks to them (the virtual experimenter) and where several other virtual human agents enter and leave the room where the participant feels present. Our manipulation leverages on the subjective experience of copresence (togetherness with others in the virtual world (Slater et al. 2022)) that occurs in such VR simulations. Although co-presence is often observed in VR with similar settings and rendering quality, our study could benefit from a direct assessment of copresence.”

We voluntarily decided not to dwell on this point any further in the main manuscript document for two main reasons. First, we did not want to dilute the core message of the paper on the translational aspect of our methods and measures regarding presence hallucination. Second, we did not want to create any confusion between the different denominations and definitions used in both fields (sense of presence (VR field) and presence hallucination - or feeling of a presence (hallucination field)).

REVIEWERS' COMMENTS

Reviewer #1 (Remarks to the Author):

My comments were mainly stylistic and the authors have dealt with them all very thoroughly. As far as I can judge, they have also dealt with the other referees' comments.

Reviewer #2 (Remarks to the Author):

The authors answered exhaustively to my previous queries, and modified the manuscript according to suggestions. I have no further questions.

Reviewer #3 (Remarks to the Author):

All my comments and methodological questions having been carefully considered and answered, I am in favor of publication, subject to the acceptance of the peer-reviewers who are experts in the disease in question.

Reviewer #3 (Remarks on code availability):

Please check folder integrity when publishing. Unless I'm mistaken, the readme file refers to a `_code` folder which is not present in the archive. The `_data` file seems coherent and the data relatively well explained, but I can't get the code, especially from the virtual environment.

Reviewer #1 (Remarks to the Author):

My comments were mainly stylistic and the authors have dealt with them all very thoroughly. As far as I can judge, they have also dealt with the other referees' comments.

We thank Reviewer #1 for the positive appreciation of our revised manuscript.

Reviewer #2 (Remarks to the Author):

The authors answered exhaustively to my previous queries, and modified the manuscript according to suggestions. I have no further questions.

We thank reviewer #2 for the positive appreciation of our revised manuscript.

Reviewer #3 (Remarks to the Author):

All my comments and methodological questions having been carefully considered and answered, I am in favor of publication, subject to the acceptance of the peer-reviewers who are experts in the disease in question.

We thank reviewer #3 for the positive appreciation of our revised manuscript.

Reviewer #3 (Remarks on code availability):

Please check folder integrity when publishing. Unless I'm mistaken, the readme file refers to a `_code` folder which is not present in the archive. The `_data` file seems coherent and the data relatively well explained, but I can't get the code, especially from the virtual environment.

Due to size constraints, the `"_code"` folder was not present in the archive submitted in the Nature manuscript tracking system. The `"_code"` folder, along with the `"_data"` folder, are now publicly available at <https://doi.org/10.5281/zenodo.10511579>.